# VPS9D1-AS1 overexpression amplifies intratumoral TGF-β signaling and promotes tumor cell escape from CD8+ T cell killing in colorectal cancer

Lei Yang[1,2]\*, Xichen Dong[1], Zheng Liu[1], Jinjing Tan[3], Xiaoxi Huang[1], Tao Wen[1], Hao Qu[2]\*, Zhenjun Wang[2]\*

[1]Medical Research Center, Beijing Chao-Yang Hospital, Capital Medical University, Beijing, China; [2]Department of General Surgery, Beijing Chao-Yang Hospital, Capital Medical University, Beijing, China; [3]Department of Cellular and Molecular Biology, Beijing Chest Hospital, Capital Medical University & Beijing Tuberculosis and Thoracic Tumor Research Institute, Beijing, China

**\*For correspondence:**
yl6649084@mail.ccmu.edu.cn (LY);
13701320206@163.com (HQ);
drzhenjun@163.com (ZW)

**Competing interest:** The authors declare that no competing interests exist.

**Abstract** Efficacy of immunotherapy is limited in patients with colorectal cancer (CRC) because high expression of tumor-derived transforming growth factor (TGF)-β pathway molecules and interferon (IFN)-stimulated genes (ISGs) promotes tumor immune evasion. Here, we identified a long noncoding RNA (lncRNA), VPS9D1-AS1, which was located in ribosomes and amplified TGF-β signaling and ISG expression. We show that high expression of VPS9D1-AS1 was negatively associated with T lymphocyte infiltration in two independent cohorts of CRC. VPS9D1-AS1 served as a scaffolding lncRNA by binding with ribosome protein S3 (RPS3) to increase the translation of TGF-β, TGFBR1, and SMAD1/5/9. VPS9D1-AS1 knockout downregulated OAS1, an ISG gene, which further reduced IFNAR1 levels in tumor cells. Conversely, tumor cells overexpressing VPS9D1-AS1 were resistant to CD8+ T cell killing and lowered IFNAR1 expression in CD8+ T cells. In a conditional overexpression mouse model, VPS9D1-AS1 enhanced tumorigenesis and suppressed the infiltration of CD8+ T cells. Treating tumor-bearing mice with antisense oligonucleotide drugs targeting VPS9D1-AS1 significantly suppressed tumor growth. Our findings indicate that the tumor-derived VPS9D1-AS1/TGF-β/ISG signaling cascade promotes tumor growth and enhances immune evasion and may thus serve as a potential therapeutic target for CRC.

## Editor's evaluation

This research work uncovered the role of a long noncoding RNA VPS9D1-AS1(VPS) in mediating immune evasion of colorectal cancer cells, which is achieved via amplifying intra-tumoral TGF-β/ISG signaling to facilitate escape from cytotoxic T cells killing. Overall, the experiments were well-designed and the data were properly analyzed. The findings are of potential significance to gaining insight into treating colorectal cancer cells by enhancing the efficacy of immunotherapy

## Introduction

Colorectal carcinoma (CRC) is a major cause of cancer-related death worldwide and shows a high propensity for metastatic dissemination (*Siegel et al., 2020*). Microsatellite-stable (MSS) CRC is regarded as immunologically 'cold', meaning that it is scarcely infiltrated by T cells and possibly nonimmunogenic and therefore unlikely to benefit from immune therapies (*Guan et al., 2021*). Thus,

immune checkpoint blockade (ICB) is more effective in microsatellite instability-high (MSI-H) CRC but not in MSS (*Liao et al., 2019*). The lack of a DNA mismatch repair mechanism in MSI patients results in a higher tumor mutation burden and thus high neoantigen exposure that favors ICB (*Lu et al., 2021*). However, important immune features, including the degree of T cell infiltration and the differentiation or activation state of T cells, remain to be elucidated (*Benci et al., 2019*).

Effective ICB relies on CD8[+] T cell infiltration in the tumor microenvironment (TME). However, advanced-stage tumor cells secrete high levels of transforming growth factor (TGF)-β to reduce the activity of intratumoral cytotoxic T lymphocytes (CTLs), thereby inhibiting their antitumor effector functions (*Katlinski et al., 2017*). As a result, solid tumors evade anticancer immunity by establishing immune-privileged niches in the TME (*Chongsathidkiet et al., 2018*). Increased TGF-β in the TME limits the adaptive immune responses by inhibiting T effector cell functions and ushering exhausted T cells to apoptosis (*Tauriello et al., 2018*; *Liu et al., 2020*). The receptors of interferon (IFN) are found to regulate TGF-β signaling pathway and are associated with CD8[+] T cell immunity (*Mariathasan et al., 2018*).

IFN signaling is essential for communication between tumor cells and their neighboring cells (*Sistigu et al., 2014*). Endogenous IFNs contribute to antitumor immunity by stimulating specific CD8α lineage dendritic cells to cross-present antigens to CTLs (*Katlinski et al., 2017*) and provide a 'third signal' to stimulate the clonal expansion of CD8[+] T cells (*Gracias et al., 2013*). In contrast, high levels of tumor-derived IFN stimulating genes (ISGs) are associated with immunological resistance. For example, IFN alpha receptor (IFNAR)–1 knockout (KO) in mouse cancer cells provoked pronounced immune responses after ionizing radiation, and the cancer cells were more susceptible to CD8[+] T cell-mediated killing (*Chen et al., 2019*). In tumor cells, PDL1 expression is promoted by IFN-γ secretion, which results in tumor cells escaping immune elimination (*Cerezo et al., 2018*). Thus, the IFN pathway plays contradictory roles in tumor cells and T lymphocytes.

VPS9D1-AS1 (also known as MYU), a long noncoding RNA (lncRNA) that has been proven to be overexpressed in multiple types of cancers (*Kawasaki et al., 2016*; *Tan and Yang, 2018*; *Wang et al., 2020*), was identified as a target of Wnt/c-Myc signaling and exhibited pro-oncogenic roles (*Kawasaki et al., 2016*). Recently, VPS9D1-AS1 was reported to enhance colon cancer progression through upregulating integrin subunit alpha 1 (*Huang et al., 2022*). VPS9D1-AS1 was demonstrated to upregulate the kinesin family member 11 through competitively sponging miRNA-30a (*Liu et al., 2021*). Here, we first report that VPS9D1-AS1 is an essential lncRNA that decreases CD8[+] T cell infiltration by enhancing TGF-β and ISG expression in CRC. In addition, we propose that VPS9D1-AS1 might serve as a drug target to enhance the efficacy of ICB treatment against CRC.

## Results

### Increased VPS9D1-AS1 levels are positively associated with TGF-β signaling in CRC tissues

To study the clinical relevance of VPS9D1-AS1 expression, we used RNAscope to evaluate VPS9D1-AS1 levels in two independent cohorts. The OUTDO cohort enrolled 158 CRC subjects, and the BJCYH cohort enrolled 49 CRC patients. The levels of VPS9D1-AS1 were significantly higher in cancer tissues than in normal intestinal epithelial tissues (*Figure 1A*, *Figure 1—figure supplement 1A*). The survival and pathological characteristic analyses demonstrated that the levels of VPS9D1-AS1 were significantly associated with overall survival (OS), TNM stage, and tumor lymph node metastasis (*Figure 1B*, *Figure 1—figure supplement 1B*). We further confirmed the overexpression (OE) of VPS9D1-AS1 in cancer tissues using qRT-PCR assays (*Figure 1C*).

Our previous study quantitatively investigated eight proteins (TGF-β, TGFBR1, TGFBR2, SMAD1/5/9, pSMAD1/5/9, SMAD2/3, pSMAD2/3, and SMAD4) involved in TGF-β signaling by multispectral fluorescence immunohistochemistry (mfIHC) staining (*Yang et al., 2018*; *Yang et al., 2019*). Because mfIHC and RNAscope assays were carried out on the same CRC tissue samples, we examined the relationships between VPS9D1-AS1 and TGF-β signaling. The protein levels of TGF-β signaling molecules were analyzed separately in the tumor and the cancer stroma. In tumor tissues, we found that CRC patients with positive VPS9D1-AS1 expression shown higher levels of TGF-β, TGFBR1, and SMAD1/5/9 (*Figure 1D-E*). We also detected the upregulation of mRNA encoding TGFBR1, SMAD1, and SMAD9 in tissue samples from BJCYH cohort using qRT-PCR (*Figure 1—figure supplement*

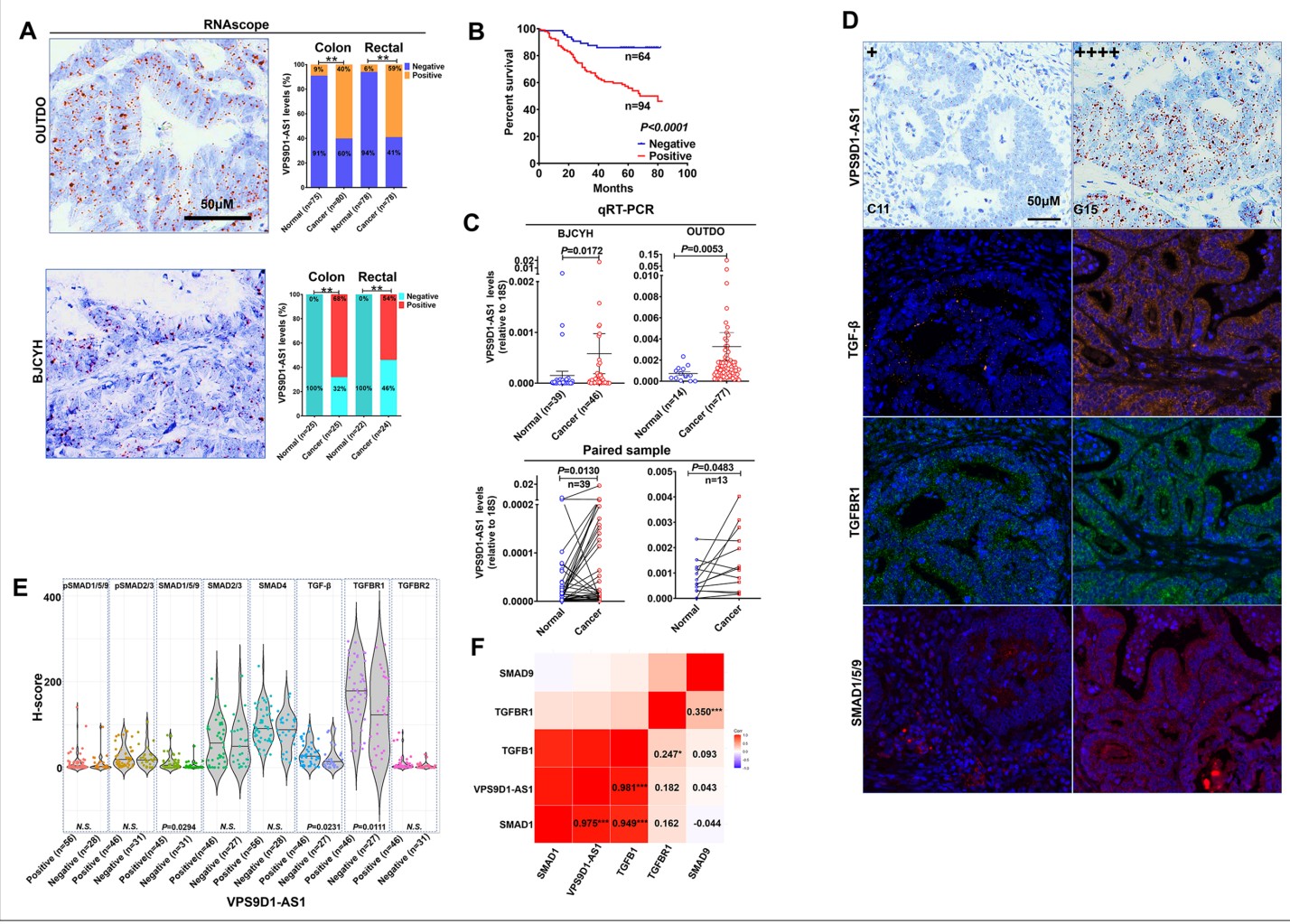

**Figure 1.** VPS9D1-AS1 is significantly upregulated in colorectal cancer (CRC) and activates the TGF-β signaling pathway. (**A**) RNAscope stained VPS9D1-AS1 in CRC tissues that were enrolled in OUTDO (upper) and BJCYH cohorts (lower). Semiquantitative analyses of the levels of VPS9D1-AS1 in cancer and normal tissues of CRC patients (right). (**B**) Kaplan-Meier overall survival curves of VPS9D1-AS1-positive and VPS9D1-AS1-negative CRC patients. (**C**) qRT-PCR evaluation of the mRNA levels of VPS9D1-AS1 (upper). Expression of VPS9D1-AS1 was compared in paired normal and cancer tissues (lower). (**D**) Representative pictures of VPS9D1-AS1-negative (+, **C11**) and VPS9D1-AS1-positive (G15, ++++) and multispectral fluorescence immunohistochemistry (mfIHC)-stained TGF-β, TGFBR1, and SMAD1/5/9 in the same CRC patients. (**E**) Integrative analysis of RNAscope and mfIHC data indicates that cancer tissues with high levels of VPS9D1-AS1 had higher levels of TGF-β, TGFBR1, and SMAD1/5/9 than these with low levels of VPS9D1-AS1. (**F**) *Pearson* correlation analyses investigated the mRNA levels of VPS9D1-AS1, TGF-β, TGFBR1, SMAD1, and SMAD9. p-Values were obtained by *chi*-square (**A**), log-rank test (**B**), unpaired *t* nonparametric test (**C, E**), paired *t* test (**C**), and *Pearson* correlation test (**F**). Data are shown as data points with mean ± standard deviation of mean (SEM) (**C**), data are depicted by violin and scatter plots with mean value (**E**). * p<0.05, ** p<0.01, *** p<0.001.

The online version of this article includes the following figure supplement(s) for figure 1:

**Figure supplement 1.** Levels of VPS9D1-AS1 were not related to TGF-β signaling in cancer stromal cells.

1C-D). In cancer stroma, VPS9D1-AS1 showed no effects on TGF-β signaling molecules (*Figure 1— figure supplement 1E*). At the mRNA level, VPS9D1-AS1 was positive associated with the levels of *TGFB1* and *SMAD1* (*Figure 1F*).

## Overexpression of VPS9D1-AS1 negatively associates with the levels of infiltrated cytotoxic T lymphocytes

To further explore the role of VPS9D1-AS1, we compared their levels in The Cancer Genome Atlas (TCGA) datasets that included consensus molecular subtype (CMS) status (*Guinney et al., 2015*). Our analyses revealed that VPS9D1-AS1 was expressed predominantly in CMS2 patients (*Figure 2—figure*

*supplement 1A*). The lymphocyte infiltration signature scores in CMS2 were significantly lower than those in CMS1, CMS3, and CMS4 (*Figure 2—figure supplement 1B*). Thus, we considered that the OE of VPS9D1-AS1 in CRC cells might be an important cause of the exclusion of T-infiltrating lymphocytes (TIL) from the TME.

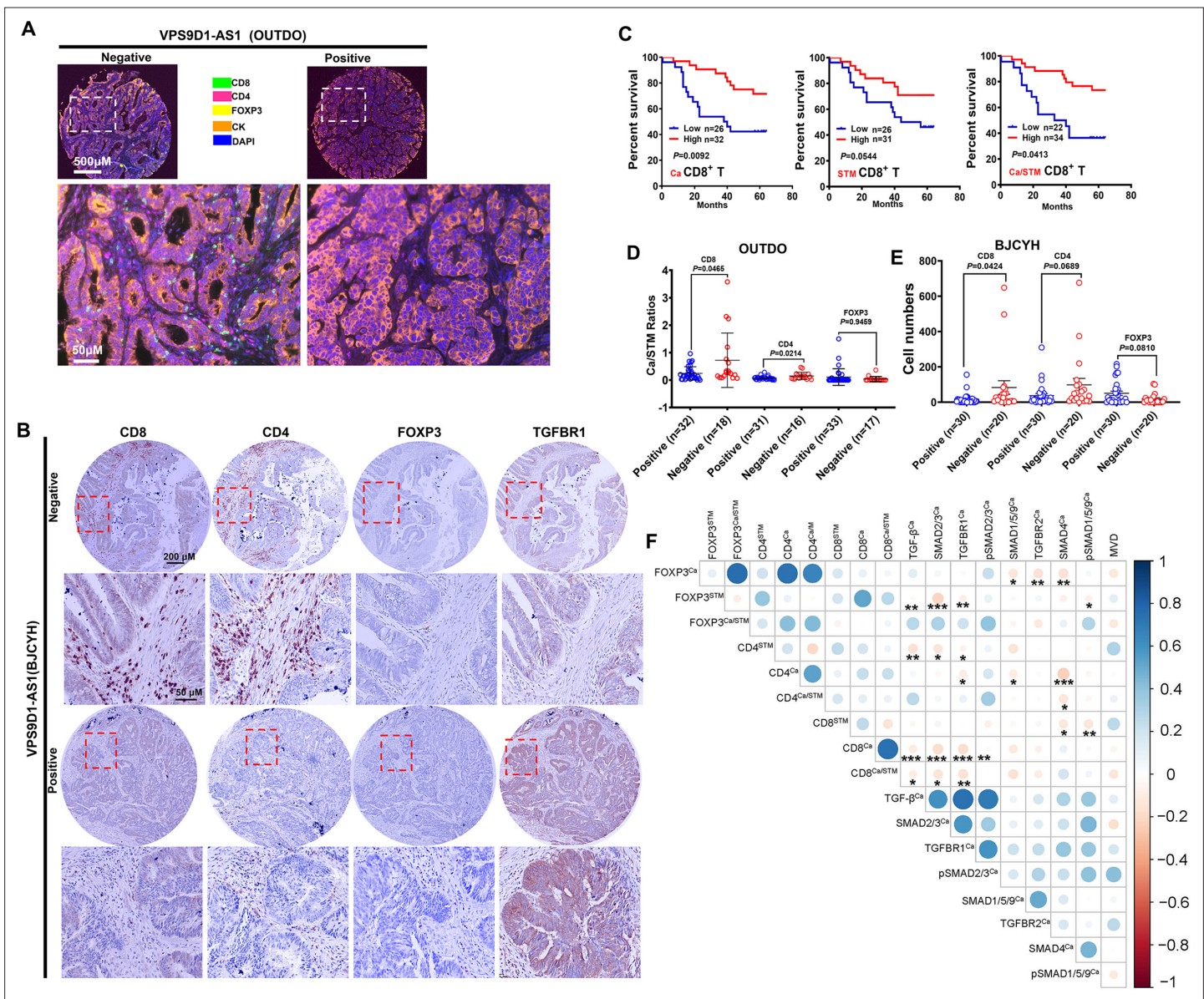

**Figure 2.** VPS9D1-AS1 is associated with reduced T lymphocyte infiltration. (**A**) Representative pictures of T cell infiltration (CD4, CD8, FOXP3) in colorectal cancer (CRC) tissues for VPS9D1-AS1 quantification. Tumor cells are marked by cytokeratin. (**B**) Representative pictures of CD8+, CD4+, FOXP3+ T cells, and TGFBR1 stained by immunohistochemistry (IHC) in the BJCYH cohort. (**C**) The overall survival curves depicting the percentage of surviving CRC patients stratified by the levels of CD8+ T cell infiltration in cancerous (Ca) tissues, cancer stroma (STM), and the Ca/STM ratio. (**D**) Ca/ STM ratios of CD4+, CD8+, and FOXP3+ T cells were calculated to identify the difference between VPS9D1-AS1 negative and positive populations in the OUTDO cohort. (**E**) The numbers of CD4+, CD8+, and FOXP3+ T cells in cancer tissues of BJCYH cohort were compared between VPS9D1-AS1 negative and positive tissues. (**F**) *Pearson* correlation analyses investigated the relationships between T-infiltrating lymphocytes and TGF-β signaling in cancer tissues. Eight protein levels were investigated by multispectral fluorescence IHC assays in same samples, and fluorescence intensity of each protein level was transferred into quantitative data for *Pearson* correlation analyses. p-Values were obtained by log-rank test (**C**), unpaired *t* nonparametric test (**D**, **E**), and *Pearson* correlation test (**F**). Data are shown by mean ± SEM (**D, E**). * p<0.05, ** p<0.01, *** p<0.001.

The online version of this article includes the following figure supplement(s) for figure 2:

**Figure supplement 1.** Integrative analysis of the relationship between VPS9D1-AS1, TGF-β signaling, and T-infiltrating lymphocytes (TILs).

To validate this hypothesis, we evaluated the levels of TILs in CRC tissue samples. In the OUTDO cohort, an mfIHC assay was carried out to calculate the percentages (%) of T lymphocytes in the total cancerous and cancer stromal cells, which represent the levels of T cell infiltration. TILs included CD4$^+$, CD8$^+$, and FOXP3$^+$ T cells, and all subsets were significantly reduced in the cancerous tissues compared to the cancer stromal tissues (*Figure 2A*, *Figure 2—figure supplement 1C*). In the BJCYH cohort, IHC assays demonstrated that the levels of CD8$^+$ T cells were decreased while FOXP3$^+$ T cells were increased in cancer tissues in comparing with matched normal tissues (*Figure 2B*, *Figure 2—figure supplement 1D*).

In the OUTDO cohort, the percentages of CD4$^+$ T and FOXP3$^+$ T cells in cancerous tissues (Ca) and cancer stroma (STM) and their ratios (Ca/STM) did not show any statistically significant relationship with OS (*Figure 2—figure supplement 1E*). In contrast, our analyses revealed that the levels of CD8$^+$ T cells in Ca and STM and the Ca/STM ratio were significantly associated with OS (*Figure 2C*). We next tried to investigate the relationship between VPS9D1-AS1 and TILs and found that the levels of TILs were not significantly different between patients with low levels of VPS9D1-AS1 and those with high levels of VPS9D1-AS1 (*Figure 2—figure supplement 1F*). Interestingly, the levels of VPS9D1-AS1 were related to the Ca/STM ratios of CD4$^+$ and CD8$^+$ T cells (*Figure 2D*), suggesting that VPS9D1-AS1 prevented T cells from entering cancer tissues. In the BJCYH cohort, the levels of CD4$^+$ and CD8$^+$ T cells were compared between VPS9D1-AS1 positive patients and these negative patients. High levels of VPS9D1-AS1 were demonstrated to associate with lower levels of TILs (CD8$^+$ T cells) (*Figure 2E*). We further performed a *Pearson* correlation analysis to explore the relationships between the levels of TGF-β signaling molecules and TILs in OUTDO cohort. In tumor cells, the protein levels of TGF-β, TGFBR1, SMAD2/3, and pSMAD2/3 were negatively associated with the Ca/STM ratio of CD8$^+$ T cells, and the levels of SMAD4 were negatively associated with the Ca/STM ratio of CD4$^+$ T cells (*Figure 2F*), which is consistent with the role of TGF-β signaling in suppressing TILs. On the other hand, the protein levels of TGF-β, SMAD2/3, TGFBR1, and pSMAD1/5/9 in cancer stromal cells were positively associated with FOXP3$^+$ T cell infiltration but negatively associated with CD8$^+$ T cell infiltration (*Figure 2—figure supplement 1G*). Together, these results suggested that high expressions of VPS9D1-AS1 were positive associated with TGF-β signaling and negative associated with the levels of infiltrated CD8$^+$ cytotoxic T cells.

## VPS9D1-AS1 is a tumor driver and positively regulates TGF-β signaling

First, we determined the levels of VPS9D1-AS1 in 16 cell lines and found that CRC cells expressed higher levels of VPS9D1-AS1 than other cells (*Figure 3A*). We designed four small guide RNAs targeting VPS9D1-AS1 (sgVPS) and used CRISPR/Cas9 to generate stable KO CRC cell lines (*Figure 3B*, *Figure 3—figure supplement 1A-B*). VPS9D1-AS1 KO significantly downregulated TGF-β, TGFBR1, and SMAD1/5/9, did not affect SMAD2/3 and SMAD4, and increased SMAD6 expression, which acts as a negative regulator of TGF-β signaling (*Figure 3C*, *Figure 3—figure supplement 1C-D*). Furthermore, inferring RNA (siRNA) was used to disrupt the expression of VPS9D1-AS1 in HCT116 cells. We confirmed that VPS9D1-AS1 knockdown (KD) decreased TGF-β, TGFBR1, and SMAD1/5/9 expression (*Figure 3—figure supplement 3E-F*). Moreover, VPS9D1-AS1 KD (both sgRNA and siRNA) had no impact on the mRNA expression of *TGFB*, *TGFBR1*, and *SMAD1*, ~5, ~9 (*Figure 3—figure supplement 2A-B*).

We next asked whether there was feedback between VPS9D1-AS1 and TGF-β signaling. Human recombinant (h)TGF-β protein and SB431542 were used to treat SW480 and HCT116 cells. These treatments had no significant effect on VPS9D1-AS1 levels in these cell lines (*Figure 3—figure supplement 2C-D*). On the other hand, the downregulation of TGF-β, TGFBR1, and SMAD1/5/9 by siRNAs reduced the levels of VPS9D1-AS1 by 40–60% compared with the controls (siNC) (*Figure 3—figure supplement 2E*). These results indicated that loss of the endogenous TGF-β signaling molecules altered the expression of VPS9D1-AS1 through a feedback loop. However, manipulating the TGF-β signaling pathway with exogenous stimuli had no effects.

We next addressed the oncogenic roles of VPS9D1-AS1 such as promoting cell proliferation, migration, and clone formation. First, we found that stable VPS9D1-AS1 KO cells exhibited morphological changes (*Figure 3D*) and decreased clone formation capacity (*Figure 3—figure supplement 2F*). VPS9D1-AS1 KO significantly inhibited cell proliferation and migration (*Figure 3E*, *Figure 3—figure supplement 2G*). Consistent with these observations, mechanistic analyses revealed that VPS9D1-AS1

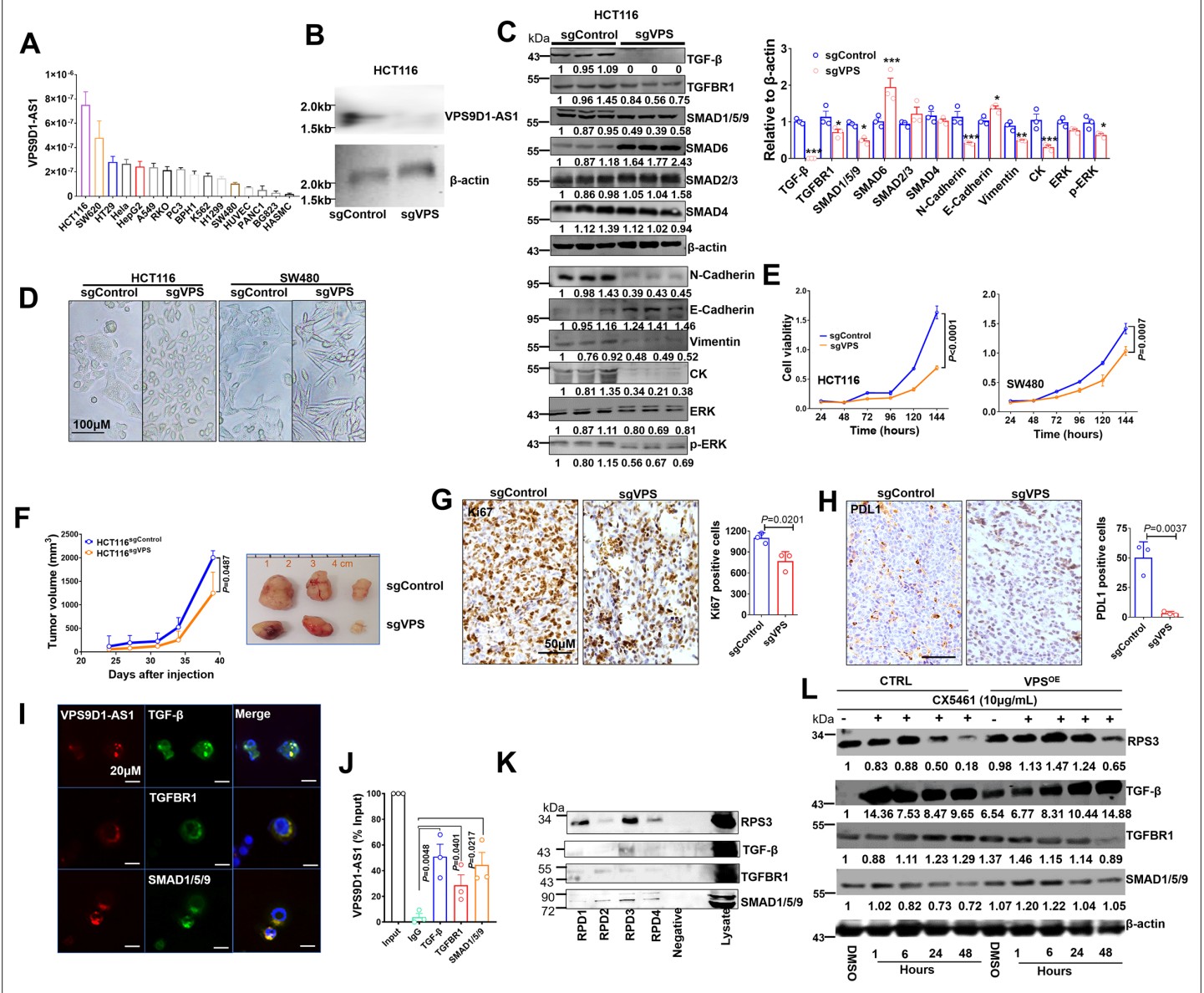

**Figure 3.** VPS9D1-AS1 controls TGF-β signaling and drives cell proliferation and metastasis. (**A**) The levels of VPS9D1-AS1 were determined by qRT-PCR in 16 cell lines. (colorectal cancer: HCT116, SW620, HT29, RKO, SW480; cervical cancer: Hela; lung cancer: A549, H1299; gastric cancer: BGC823; prostatic cancer: PC3, BPH1; leukemia: K562; pancreatic cancer: PANC1; live cancer: HepG2; HUVEC: human umbilical vein endothelial cell; HASMC: human atrial smooth muscle cell). (**B**) Northern blotting validated the knockout (KO) of VPS9D1-AS1. (**C**) Western blotting measured the levels of proteins involved in TGF-β, EMT, and ERK signaling pathways. (**D**) Representative pictures show the cell morphologies of HCT116 and SW480 cell lines. (**E**) The proliferation of HCT116/SW480 sgControl and sgVPS cells was determined by Cell counting kit-8 (CCK8) assays. (**F**) Proliferation of xenograft tumors derived from HCT116 sgControl and sgVPS cells. (**G**) Immunohistochemistry determined the levels of Ki67 and (**H**) PDL1 in xenograft tissues. (**I**) RNA fluorescence in situ hybridization (FISH)-immunofluorescence (IF) and (**J**) RNA immunoprecipitation (RIP) assays showed the interaction between VPS9D1-AS1 and proteins that included TGF-β, TGFBR1, and SMAD1/5/9. (**K**) RNA pulldown-Western blotting assays detected the interaction between VPS9D1-AS1 and the intended proteins. (**L**) Western blotting determined the changes in RPS3, TGF-β, TGFBR1, and SMAD1/5/9 in HCT116 control (CTRL) and VPS9D1-AS1 (VPS)-overexpressing (OE) cells treated with CX5461. RPD, RNA pull down probe. p-Values were obtained by two-way ANOVA (**E, F**) and paired or unpaired *t* tests (**C, G, H, J**). Data are shown as the mean ± SEM (**C, E, F, G, H, J**). *p<0.05, ** p<0.01, *** p<0.001.

The online version of this article includes the following source data and figure supplement(s) for figure 3:

**Source data 1.** TGF-β, TGFBR1, SMAD4, SMAD1/5/9, SMAD6, SMAD2/3, β-actin, N-cadherin, E-cadherin, vimentin, CK, ERK, and p-ERK western blot for ***Figure 3C***.

**Source data 2.** RPS3 RNA-pull-down western blot for ***Figure 3K***.

**Source data 3.** SMAD1/5/9, RPS3, TGF-β, and TGFBR1 western blot for ***Figure 3L***.

*Figure 3 continued on next page*

*Figure 3 continued*

**Figure supplement 1.** VPS9D1-AS1 activated TGF-β signaling.

**Figure supplement 2.** VPS9D1-AS1 regulated TGF-β signaling and promoted tumor proliferation and migration.

**Figure supplement 3.** VPS9D1-AS1 functions as the scaffolding lncRNA.

KO reduced the levels of ERK, pERK, N-cadherin, vimentin, and cytokeratin but increased the level of E-cadherin (*Figure 3C*, *Figure 3—figure supplement 2H*). In xenograft models, VPS9D1-AS1 KO significantly reduced the tumor volumes compared with the controls and significantly decreased the Ki67 and PDL1 levels in xenograft tumors (*Figure 3F-H*, *Figure 3—figure supplement 2I*). Specifically, SW480 VPS9D1-AS1 KO cells did not form xenograft tumors in mice (*Figure 3—figure supplement 2I*). Taken together, these findings support the notion that VPS9D1-AS1 acts as the driver of tumor progression by activating the ERK and EMT pathways.

## VPS9D1-AS1 scaffolds TGF-β signaling-related proteins

We predicted that VPS9D1-AS1 might act as scaffolding lncRNA in tumor cells. To validate this hypothesis, RNA FISH-immunofluorescence (IF) assays were conducted, and the colocalization of VPS9D1-AS1, TGF-β, TGFBR1, and SMAD1/5/9 was confirmed in SW480 cells (*Figure 3I*). RNA immunoprecipitation (RIP) assays further showed that VPS9D1-AS1 directly bound to TGF-β, TGFBR1, and SMAD1/5/9 (*Figure 3J*).

We next predicted the subcellular localization of VPS9D1-AS1 by lncLocator (*Lin et al., 2021*) and found that most transcripts of VPS9D1-AS1 were localized in ribosomes (*Figure 3—figure supplement 3B*). To map the protein binding regions in VPS9D1-AS1, we synthesized four biotinylated RNA probes targeting VPS9D1-AS1 transcript (*Figure 3—figure supplement 3A*). Our RNA pulldown (RPD) assay also proved that VPS9D1-AS1 bound with RPS3, one of the proteins constituting the small ribosomal subunit (*Figure 3K*). Thus, we sought to determine whether preventing ribosome biogenesis plays a role in regulating the translation of TGF-β, TGFBR1, and SMAD1/5/9 (*Devlin et al., 2016*). We found that VPS9D1-AS1 OE prevented RPS3 degradation caused by CX5461 (an inhibitor of RNA polymerase I transcription of ribosomal RNA genes) treatment. However, CX5461 treatment immediately increased the levels of TGF-β, which declined over time. In VPS9D1-AS1 OE cells, the levels of TGFBR1 and SMAD1/5/9 were decreased after treatment with CX5461, but the TGF-β levels did not decrease following the degradation of RPS3 (*Figure 3L*). These findings suggest that VPS9D1-AS1 scaffolds the TGF-β protein and regulates its translation in ribosomes.

## IFN signaling activation induced by VPS9D1-AS1 expression acts downstream of TGF-β signaling

RNA sequencing was performed to identify the mRNAs differentially expressed between HCT116 sgControl and sgVPS cells. A total of 705 differentially expressed genes were identified, which included 203 upregulated genes and 502 downregulated genes (*Figure 4A*, *Figure 4—figure supplement 1A*). VPS9D1-AS1 KO significantly inactivated IFNα/β signaling and the cell death pathway as well as immune system processes (*Figure 4B*, *Figure 4—figure supplement 1B-C*). Seventeen genes involved in IFNα/β signaling were validated in HCT116, RKO, and SW480 cells. IFI27 and OAS1 were the most significantly downregulated genes upon VPS9D1-AS1 KO (*Figure 4C*, *Figure 4—figure supplement 1D*). Analyses in TCGA datasets demonstrated that OAS1 and IFI27 were significantly overexpressed in CRC cancer tissues (*Figure 4—figure supplement 1E*). We also analyzed the mRNA expression of OAS1 and IFI27 in tissue samples from 26 CRC cancer tissues and 10 normal colon tissues (*Figure 4D*). *Pearson* correlation analysis revealed that the levels of OAS1, but not IFI27, were significantly related to VPS9D1-AS1 levels (*Figure 4E*).

STAT1 is a well-known transcription factor activated by various ligands, including IFNα (*Cerezo et al., 2018*). After hIFNα stimulation, we found that VPS9D1-AS1 KO resulted in the downregulation of STAT1 and pSTAT1 (*Figure 4F*). When the cells were treated with hTGF-β, the phosphorylation of STAT1 induced by hIFNα stimulation was inhibited (*Figure 4—figure supplement 1F*). We further performed siRNA-mediated KD of TGF-β, TGFBR1, and SMAD1 and confirmed that blocking TGF-β signaling reduced OAS1 and IFI27 expression (*Figure 4G*). The expression levels of OAS1 were more significantly affected than IFI27 (*Figure 4G*). In contrast, VPS9D1-AS1 OE restored OAS1 and

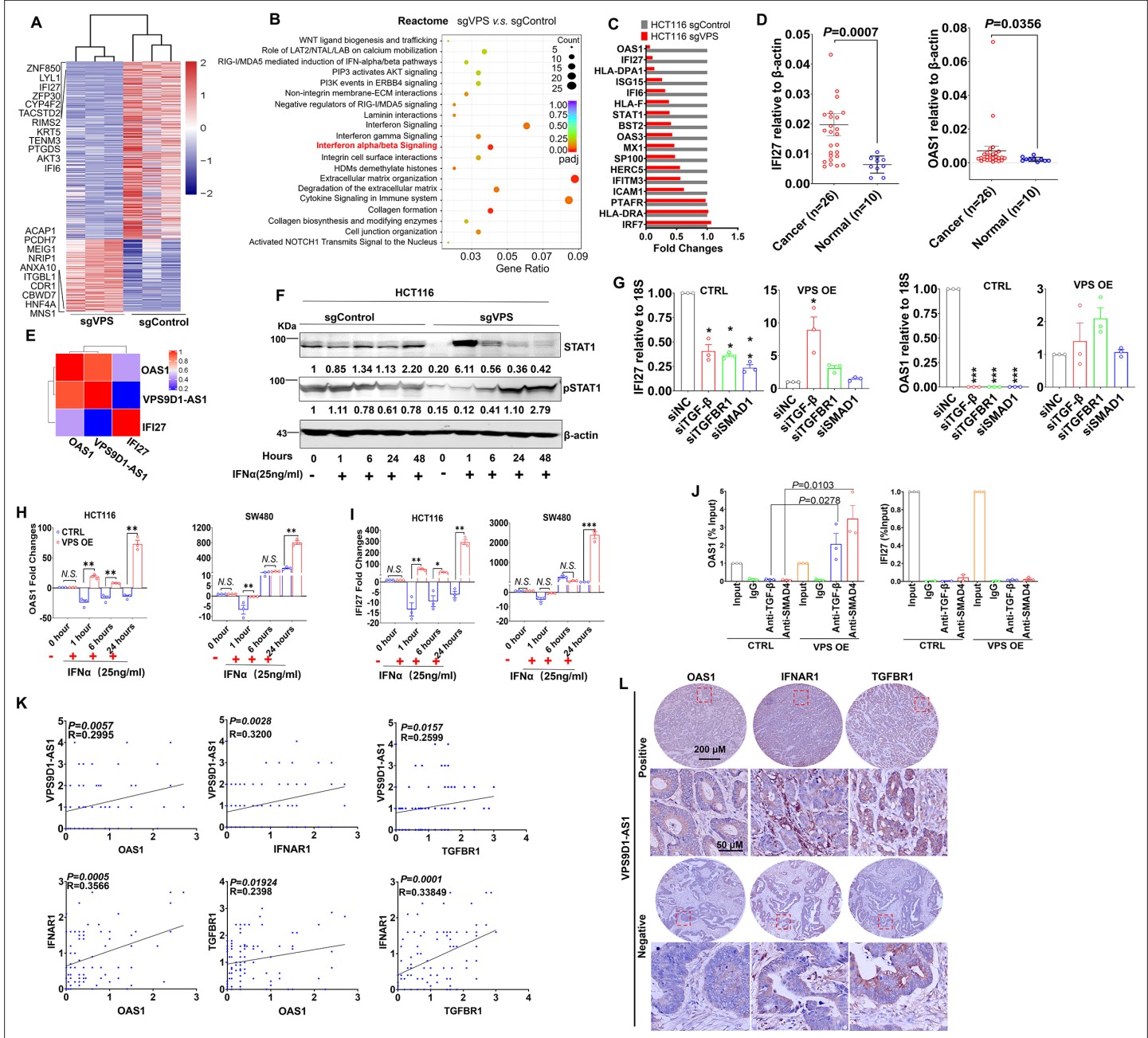

**Figure 4.** VPS9D1-AS1 regulates interferon signaling. (**A**) Heatmap illustrating the results of RNA sequencing of the genes regulated by VPS9D1-AS1. (**B**) VPS9D1-AS1 regulated the pathways associated with interferon signaling. (**C**) Differential expression of 17 genes in the IFNα/β signaling pathway was validated in HCT116 cells. (**D**) Colorectal cancer (CRC) tissue mRNA levels of IFI27 and OAS1 were determined by qRT-PCR, and (**E**) their relationships with VPS9D1-AS1 were calculated with *Pearson* correlation analysis. (**F**) Effect of VPS9D1-AS1 on STAT1 pathway activation induced by human recombinant IFNα. (**G**) VPS9D1-AS1 overexpression (OE) increased the expression of IFI27 and OAS1 through activated TGF-β signaling. (**H**) OAS1 and (**I**) IFI27 levels exhibited disparate changes upon IFNα stimulation in VPS9D1-AS1 OE cells and control (CTRL) cells. (**J**) Chromatin immunoprecipitation (ChIP) assays demonstrated the interactions between SMAD4 and the promoter regions of OAS1. (**K**) *Pearson* correlation analyses investigated the relationships among VPS9D1-AS1, OAS1, IFNAR1, and TGFBR1 in CRC tissues. (**L**) Immunohistochemistry assays showed the levels of OAS1, IFNAR1, and TGFBR1 in patients with negative or positive expression of VPS9D1-AS1. p-Values were obtained by unpaired t test (**D, J**), two-way ANOVA (**G, H, I**), and *Pearson* correlation (**K**). Data are shown as the mean ± SEM (**D, G, H, I, J**). * p<0.05, ** p<0.01, *** p<0.001. *N.S.*, not significant.

The online version of this article includes the following source data and figure supplement(s) for figure 4:

**Source data 1.** STAT1 and pSTAT1 western blot for *Figure 4F*.

**Figure supplement 1.** VPS9D1-AS1 plays a role on interferon signaling.

IFI27 expression (*Figure 4G*). Surprisingly, OAS1 and IFI27 mRNA levels were significantly reduced in VPS9D1-AS1 OE cells, although VPS9D1-AS1 was stably upregulated by ~180.40 times in HCT116 cells and ~42.28 times in SW480 cells (*Figure 4—figure supplement 1G-H*). However, hIFNα stimulation significantly increased OAS1 and IFI27 in VPS9D1-AS1 OE cells (*Figure 4H-I*). These results led us to hypothesize that OAS1 activated IFN signaling that was dependent on VPS9D1-AS1.

SMAD4 enters the nucleus and acts as transcription factors and induced transcription factor which regulates gene expression (*Derynck et al., 2021*). We confirmed that SMAD4 antibodies immunoprecipitated the promoter regions of both the *OAS1* (–77 to +284) and *IFI27* (–1933 to –1843) genes (*Figure 4—figure supplement 1I*). Importantly, VPS9D1-AS1 OE significantly enhanced the binding between SMAD4 and the promoter regions of *OAS1* but failed to enhance this binding in the *IFI27* promoter region (*Figure 4J*). Moreover, IHC analysis indicated that the levels of OAS1, IFNAR1, and TGFBR1 were consistently elevated in CRC tissue samples that had positive VPS9D1-AS1 expression (*Figure 4K*~L). Collectively, OAS1 involved in IFN signaling acts downstream of TGF-β signaling and increased by VPS9D1-AS1 OE.

## VPS9D1-AS1 mediates the crosstalk between tumors and T cells depending on IFNAR1 expression

The receptor for IFNα/β is one of the downstream targets of ISGs. When we examined the changes in IFNAR1 on the surface of HCT116 cells by flow cytometry (FCM), we found that VPS9D1-AS1 KO reduced the expression of IFNAR1 (*Figure 5A*). Conversely, VPS9D1-AS1 OE significantly upregulated the expression of IFNAR1 in tumor cells (*Figure 5B*). To further delineate the upstream pathway essential for IFNAR1 upregulation in tumor cells, we applied the CRISPR/Cas9 technique to abolish OAS1 expression (*Figure 5—figure supplement 1A*). Our results demonstrated that the deletion of OAS1 decreased the expression of IFNAR1, although VPS9D1-AS1 OE cells were resistant to this effect (*Figure 5B*). In addition, the deletion of OAS1 inhibited STAT1 and TGF-β expression (*Figure 5—figure supplement 1A*).

To investigate the interaction of T cells and tumor cells, we prepared T cells from the peripheral blood of healthy donors. CD8+ T cells and total lymphocytes were separately cultured and primed with human interleukin 2 and antibodies against CD3 and CD28. In vitro-primed T cells were cultured with HCT116 sgControl and sgVPS cell lines. Coculture with VPS9D1-AS1 KO cells increased IFNAR1 expression in CD8+ and CD4+ T cells (*Figure 5C*, *Figure 5—figure supplement 1B*). We further detected the CD4+ IFNAR1 levels and CD8+ PD1 levels of peripheral blood lymphocytes in 15 patients with VPS9D1-AS1 positive expression and 16 patients with VPS9D1-AS1 negative expression in cancer tissues (*Figure 5D-E*). Although there was no statistically significant difference, we found that CRC patients with tissue positive VPS9D1-AS1 expression shown lower levels of IFNAR1 in CD4+ T cells. However, VPS9D1-AS1 was found to be positively associated with the expression levels of PD1 of peripheral blood CD8+ T cells.

We further developed a T cell cytotoxicity assay using MC38-OVA (ovalbumin) cells. Our models successfully demonstrated that primed CD8+ T cells from OT-1 mice suppressed the proliferation of MC38-CTRL-OVA and MC38-VPS9D1 OE-OVA tumor cells (*Figure 5F*, *Figure 5—figure supplement 1C*). Although there were no statistically significant differences, we observed that supplied anti-TGF-β and anti-PD1 antibodies in media with OT-1 cells enhanced the cytotoxicity of CD8+ T cells in killing VPS9D1-AS1 OE cells (*Figure 5F*). FCM analysis showed that CD8+ T cells secreted more IFN-γ once they contacted tumor cells. However, VPS9D1-AS1 OE tumor cells inhibited CD8+ T cells from secreting IFN-γ. Furthermore, neutralizing antibodies against PD1 restored IFN-γ secretion by CD8+ T cells (*Figure 5G*).

These data support the idea that VPS9D1-AS1 upregulates OAS1 by enhancing TGF-β signaling derived from cancer cells to protect themselves from T cell-mediated cytotoxicity through regulation of IFNAR1.

## Upregulation of VPS9D1-AS1 inhibits antitumor immune cell infiltrations in immune-competent mice

The human *VPS9D1-AS1* gene is located on the plus strand, while the protein coding gene *VPS9D1* is located on the minus strand of human chromosome 16. NR045849 is a mouse lncRNA located on the plus strand near the *Vps9d1* gene (*Figure 6—figure supplement 1A*). When we ectopically

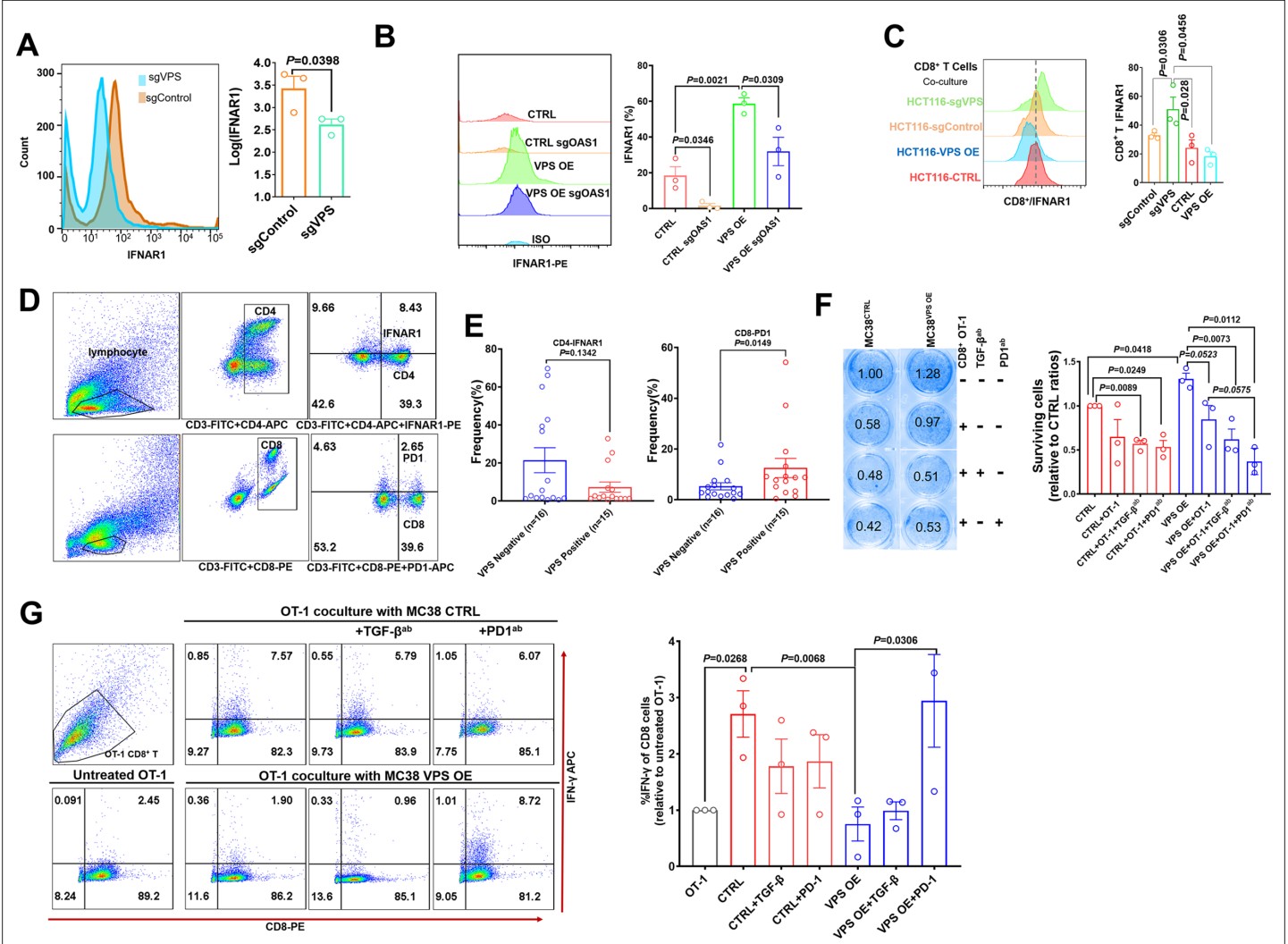

**Figure 5.** VPS9D1-AS1 mediates crosstalk between T cells and cancer cells by regulating IFNAR1. (**A**) Flow cytometry (FCM) revealed the decrease in IFNAR1 after VPS9D1-AS1 knockout in HCT116 cells and (**B**) in HCT116 sgControl and sgVPS cells after CRISPR/Cas9-mediated inhibition of OAS1. Experiments were repeated three times. (**C**) Levels of IFNAR1 on the surface of CD8[+] T cells cocultured with HCT116 control (CTRL), VPS overexpression (OE), sgControl, and sgVPS cells were detected by FCM in three independent assays. (**D**), (**E**) FCM determination demonstrated that colorectal cancer patients with positive VPS9D1-AS1 expression had lower levels of IFNAR1 in CD4[+] and higher levels of PD1 in CD8[+] T cells in peripheral blood than these patients with negative VPS9D1-AS1 expression. (**F**) T cell cytotoxicity assays against CTRL and VPS9D1-AS1-OE MC38-OVA cell lines by OT-1 CD8[+] T cells. VPS9D1-AS1 OE reduced the cytotoxicity of activated OT-1 CD8[+] T cells. (**G**) FCM determined the IFN-γ levels in OT-1 CD8[+] T cells after exposure to CTRL and VPS-OE MC38 cells. Antibodies against TGF-β and PD1 were added to the medium. The experiments were repeated three times. p-Values were obtained by unpaired t test (**A, B, C, E, F, G**). Data are shown as the mean ± SEM (**A, B, C, E, F, G**).

The online version of this article includes the following figure supplement(s) for figure 5:

**Figure supplement 1.** VPS9D1-AS1 mediates the crosstalk between tumors and T cells through TGF-β and IFN signaling.

expressed full-length VPS9D1-AS1 in MC38 and CT26W cell lines (*Figure 6—figure supplement 1B*), we observed an increase in the expression levels of TGF-β, TGFBR1, SMAD1, and STAT1 (*Figure 6A*). NR045849 OE did not increase the levels of TGF-β in MC38 cells (*Figure 6A*). These findings indicated that VPS9D1-AS1 shared similar biological functions in murine and human cells. Thus, we decided to explore the roles of VPS9D1-AS1 in vivo by OE in murine tumor cells.

We further upregulated VPS9D1-AS1 more than 600-fold through 3 transfection cycles of lenti-VSP9D1-AS1 in MC38 cells. VPS9D1-AS1 OE cells exhibited a higher speed of growth than control cells in vitro (*Figure 6B*). VPS9D1-AS1 OE MC38 cells as well as CTRL cells were subcutaneously injected into C57BL/6 mice (*Figure 6—figure supplement 1C*). MC38 VPS OE cell-derived tumors

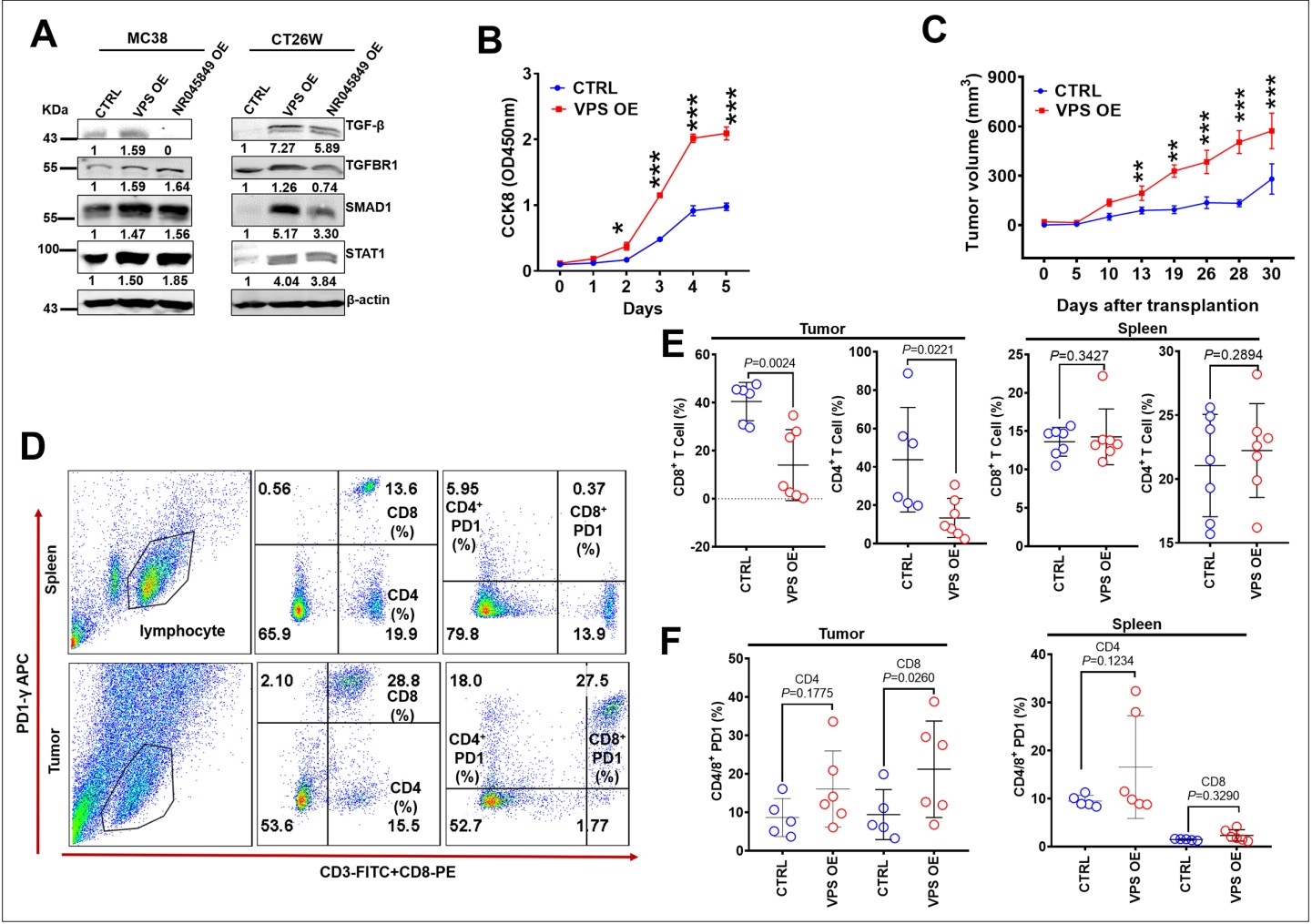

**Figure 6.** VPS9D1-AS1 overexpression (OE) cells inhibited T cell function in vivo. (**A**) VPS9D1-AS1 OE promoted the expression of TGF-β, TGFBR1, SMAD1, and STAT1 in murine cells. (**B**) MC38 VPS9D1-AS1 OE and control (CTRL) cells were determined proliferation using CCK8 assays. (**C**) Growth curves of MC38 allograft tumors (n=7 per group). (**D**) Plots represent the percentages of CD8+ and CD4+ T cells and PD1 frequencies in allograft tumors and spleens. (**E**) Levels of CD8+ and CD4+ T cells were compared in the tumor and spleen. (**F**) Levels of PD1 in CD4+ T and CD8+ T cells were compared between CTRL and VPS OE allograft tumors (left) and spleens (right). p-Values were obtained by two-way ANOVA (**B**) and unpaired t nonparametric tests (**E**, **F**). Data are shown as the mean ± SEM (**E**, **F**). **p<0.01, ***p<0.001.

The online version of this article includes the following source data and figure supplement(s) for figure 6:

**Source data 1.** TGF-β, SMAD1, STAT1, TGFBR1, and β-actin western blot for **Figure 6A**.

**Figure supplement 1.** VPS9D1-AS1 inhibited T-infiltrating lymphocyte in allograft tumors.

grew faster than MC38 CTRL cell-derived tumors (**Figure 6C**, **Figure 6—figure supplement 1D-F**). We also investigated T cells in mouse spleens and infiltrating T cells in tumor allograft tissues (**Figure 6D**). The splenic CD8+ T and CD4+ T cells showed no significant differences between mice injected with MC38 VPS OE and CTRL cells (**Figure 6E**). When allograft tumors were digested into single cells and analyzed by FCM, however, we found that VPS9D1-AS1 OE prevented CD4/8+ T cell from infiltrating into the tumor tissue (**Figure 6E**). These data underscore the inhibitory roles of VPS9D1-AS1 in T cell tumor infiltration.

We further assessed T cell surface markers that included CD44, CD107, CD62L, CCR7, and PD1 (**Figure 6E**, **Figure 6—figure supplement 1G**). The levels of PD1 on CD8+ T cells were elevated in mice bearing MC38 VPS OE tumors (**Figure 6F**), while CD44, CD107, CD62L, and CCR7 expression were not significantly different in either spleens or allografted tumor tissues (**Figure 6—figure supplement 1H**). These results indicated that VPS9D1-AS1 regulated the pathway that controls the activation or differentiation of CD8+ T cells.

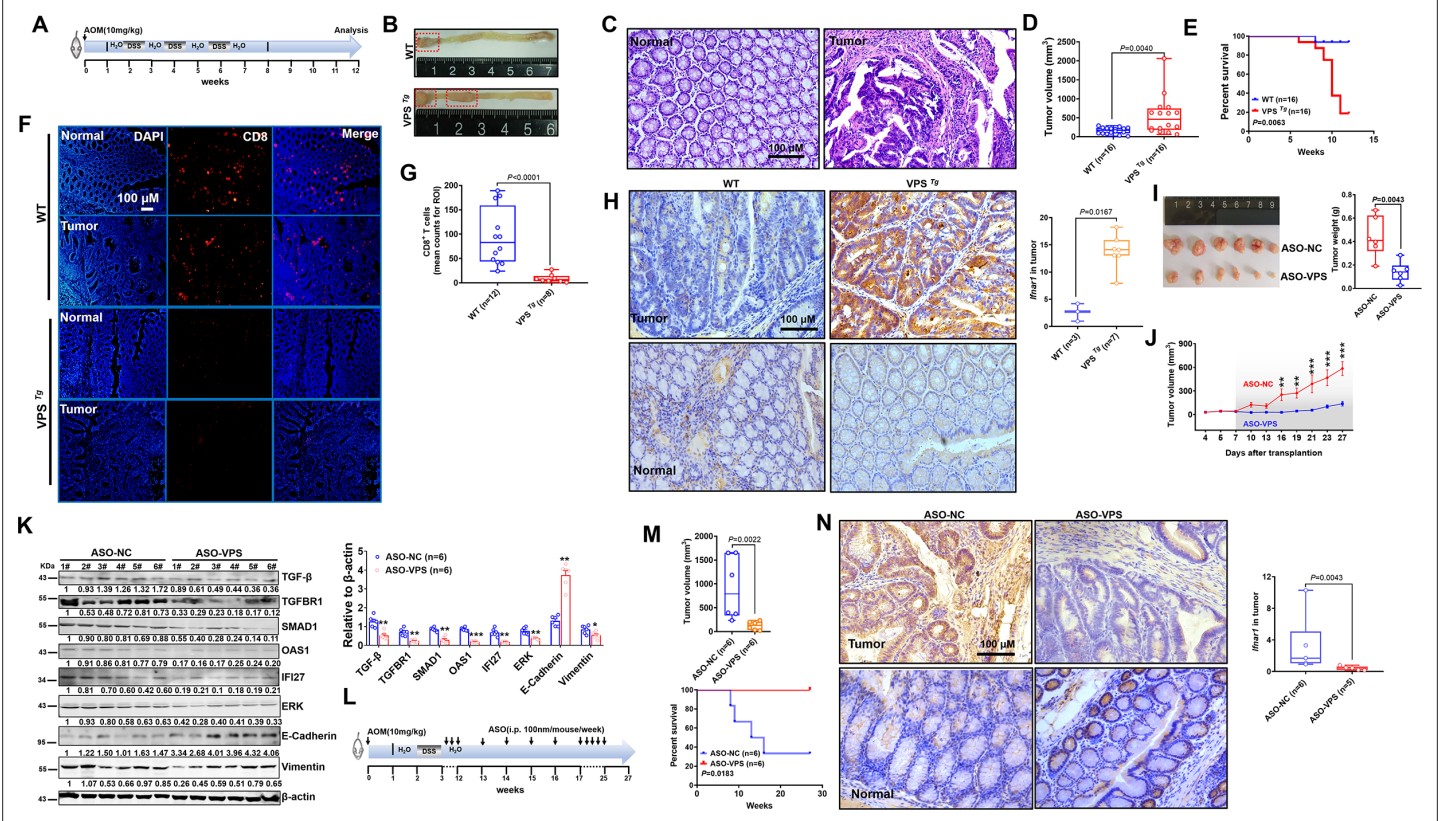

**Figure 7.** Transgenic mice validated the inhibitory role of VPS9D1-AS1 on CD8⁺ T cell infiltration and demonstrated the antitumor effects of targeting VPS9D1-AS1 with an antisense oligonucleotide (ASO) drug. (**A**) Treatment scheme of the azoxymethane/dextran sulfate sodium salt (AOM/DSS) colorectal cancer (CRC) model. Endpoint at 12 weeks. (**B**) Representative images of colorectal tumors in mice of the indicated genotypes. (**C**) Hematoxylin (H) and eosin (E) staining showing the AOM/DSS-induced murine tumors. (**D**) Tumor volumes are plotted for wild-type (WT) and VPS *Tg* mice. (**E**) The overall survival (OS) curve depicts the difference in survival rate for WT and VPS *Tg* mice. (**F**), (**G**) VPS9D1-AS1 suppressed CD8⁺ T cell infiltration in AOM/DSS-induced CRC tissues. (ROI, region of interest). (**H**) The levels of Ifnar1 were upregulated in VPS *Tg* tumors compared with WT tumors. (**I**) Tumors, their weights, and (**J**) growth curves of HCT116 xenograft tumors after injecting ASO-VPS (n=6) and ASO-NC (n=6). (**K**) Western blotting assays of the proteins involved in TGF-β, IFN, ERK, and EMT signaling in xenograft tumors. (**L**) Treatment scheme of ASO-treated mice with AOM/DSS-induced CRC. (**M**) ASO-VPS treatment decreased tumor volumes and increased OS of VPS *Tg* mice with AOM/DSS-induced CRC. (**N**) Ifnar1 expression upon ASO-VPS and ASO-NC treatment. p-Values were obtained by unpaired t nonparametric test (**D, G, H, I, K, M, N**), log-rank test (**E, M**), and two-way ANOVA tests (**J**). Data are shown as the mean ± SEM (**D, G, H, I, K, M, N**). * p<0.05, ** p<0.01, *** p<0.001.

The online version of this article includes the following source data and figure supplement(s) for figure 7:

**Source data 1.** TGF-β, TGFBR1, SMAD1, OAS1, IFI27, ERK, E-cadherin, vimentin, and β-actin western blot for *Figure 7K*.

**Figure supplement 1.** VPS9D1-AS1 drives azoxymethane/dextran sulfate sodium salt (AOM/DSS)-induced mouse model of colorectal cancer.

**Figure supplement 2.** IFNAR1 knockdown (KD) inhibits tumor proliferation in VPS9D1-AS1 overexpression (OE) cells.

**Figure supplement 3.** Illustration of the mechanism of VPS9D1-AS1 on promoting immune evasion via TGF-β signaling and interferon (IFN)-stimulated genes (OAS1 and IFNAR1).

## VPS9D1-AS1 promotes tumorigenesis in AOM/DSS-induced intestinal cancer and acts as a therapeutic target

Using the VPS9D1-AS1 transgenic (VPS *Tg*) mouse model, we generated an azoxymethane/dextran sulfate sodium salt (AOM/DSS) model to further address the oncogenic roles of VPS9D1-AS1 in vivo (*Figure 7—figure supplement 1A*). VPS9D1-AS1 was found to specifically express in intestinal epithelium of VPS *Tg* mice (*Figure 7—figure supplement 1B*). VPS *Tg* and wild-type (WT) mice from the same founder were subjected to AOM/DSS treatment cycles (*Figure 7A*). After AOM/DSS treatment, VPS *Tg* mice showed markedly higher intestinal tumor growth than WT mice (*Figure 7B-D*, *Figure 7—figure supplement 1C*). The VPS *Tg* mice had a shorter OS time than the WT mice (*Figure 7E*). Interestingly, VPS *Tg* mice showed lower CD8⁺ T cell infiltration (*Figure 7F-G*). We further determined the levels

of Ifnar1 in murine tumors and found that there were no differences in Ifnar1 levels between cancer and normal tissues (*Figure 7—figure supplement 1D*). However, Ifnar1 levels in tumor tissues were increased in VPS $^{Tg}$ mice (*Figure 7H*), indicating that VPS9D1-AS1 OE promoting Ifnar1 expression in tumor cells enhanced tumor growth.

To address the therapeutic potential of targeting VPS9D1-AS1, three 2-O-methyl antisense oligo-nucleotides (ASOs) specifically targeting VPS9D1-AS1 and one negative control were developed. Transfection of ASOs targeting VPS9D1-AS1 impaired the proliferation of HCT116, SW480, and MC38 VPS OE cells in culture (*Figure 7—figure supplement 1E*). Among them, ASO-siVPS3 showed the highest inhibition efficiency and was further used in an in vivo assay. We treated BALB/c nude mice bearing HCT116 xenograft tumors with intratumoral injection of ASO drugs and demonstrated that ASO-VPS significantly inhibited tumor growth in vivo (*Figure 7I~J*, *Figure 7—figure supplement 1F*). Moreover, the levels of TGF-β, TGFBR1, SMAD1, OAS1, IFI27, ERK, N-cadherin, and vimentin were decreased, while E-cadherin was increased in ASO-VPS-treated xenograft tumors (*Figure 7K*).

VPS $^{Tg}$ mice were subjected to the AOM/DSS cycle to investigate whether VPS9D1-AS1 acts as a therapeutic target (*Figure 7L*, *Figure 7—figure supplement 1G*). At the 12th week, mice were divided into two groups and treated with ASO-NC and ASO-VPS drugs. In the ASO-NC-treated group, four (4/6) mice died because of aggressive tumor progression in the intestinal tract (*Figure 7M*). In contrast, all six mice treated with ASO-VPS survived until the final follow-up at the 27th week. Among these six mice, five mice developed CRC, but their body weights were heavier than those of ASO-NC-treated mice from the 12th week to the 27th week (*Figure 7—figure supplement 1H*). ASO-VPS significantly reduced the tumor volumes (*Figure 7M*) and reduced Ifnar1 expression in murine tumors compared to ASO-NC treatment (*Figure 7N*). To investigate the function of IFNAR1 on VPS9D1-AS1 promoting cell proliferation, stable cell lines with IFNAR1 KD were produced in VPS9D1-AS1 OE cells and CTRL cells (*Figure 7—figure supplement 2A*). The IFNAR1 KD significantly inhibited HCT116 VPS OE cell proliferation (*Figure 7—figure supplement 2B*). Collectively, our in vivo analyses indicated that VPS9D1-AS1 was the driver of CRC, inhibited CD8+ T cell infiltration, and could serve as a therapeutic target.

## Discussion

Our present study shows that VPS9D1-AS1 synergizes with the TGF-β and IFN signaling pathways to form a signaling axis in which high intratumor cell activation prevents CTL evasion in the TME. Mechanistically, VPS9D1-AS1 enhances TGF-β signaling by increasing the expression of TGF-β, TGFBR1, and SMAD1/5/9 at the protein level through scaffold mechanisms. Meanwhile, VPS9D1-AS1 KO in tumor cells downregulates the expression of many ISG genes, thus inactivating IFN signaling, and VPS9D1-AS1 KO in tumor cells increases the sensitivity of tumor cells to T cell cytotoxicity, which is illustrated by high levels of secreted IFN. Thus, VPS9D1-AS1/TGF-β/ISG signaling consists of the pathway that crosstalks between tumors and CD8+ T cells. Our in vivo assays support the use of VPS9D1-AS1 as a therapeutic target for CRC treatment (*Figure 7—figure supplement 3*).

The dual roles of IFN signaling in tumor immune reactions have recently received intense attention (*Benci et al., 2016*). Autocrine type I IFNs from activated T cells augment T cell-mediated tumor regression (*Chen et al., 2021*) and are essential for the proliferation and differentiation of T cells (*Lopes et al., 2021*). In the TME, both tumor cells and infiltrating T cells express IFNA/GR, which sense and bind IFN molecules and then activate the expression of a series of ISGs (*Grasso et al., 2021*; *Jiang et al., 2018*). On the other hand, the suppression of IFN signaling in tumor cells stimulates the production of chemokines, such as CXCL13, which results in tumor infiltration of NK cells and subsequent inhibition of tumor progression (*Muthalagu et al., 2020*). Our data show that VPS9D1-AS1 OE in CRC cells increases IFNAR1 through upregulated OAS1, which is an ISG gene. Hundreds of ISG genes have been identified, and they act either as tumor drivers, such as ADAR1 (*Fritzell et al., 2019*), or tumor repressors, such as UBA7 (*Fan et al., 2020*). We also found that both OAS1 and IFI27 are ISGs highly expressed in CRC. Consistent with these findings, OAS1 is a prognostic factor in breast cancer (*Zhang and Yu, 2020*), while IFI27 is overexpressed in cholangiocarcinoma (*Chiang et al., 2019*). In addition, OAS1 (but not IFI27) is the downstream ISG gene that is activated by VPS9D1-AS1.

IFN-activated STAT1 promotes PDL1 expression in tumors, which further accelerates tumor progression (*Lv et al., 2021*). This result is consistent with our finding that STAT1 is activated by VPS9D1-AS1. However, the effect of VPS9D1-AS1 on promoting IFN-induced phosphorylation of STAT1 does not

persist for a long time, and it might be associated with the role of pSTAT1 in activating the downstream cGAS-STING pathway, which is essential for defective-mismatch-repair-mediated immunotherapy (*Guan et al., 2021*).

Previous work has shown that TGF-β blocks IFN production, which is secreted in a paracrine fashion by activated CTLs and limits tumor regression (*Guerin et al., 2019*). Antibodies reduce TGF-β signaling in cancer stromal cells, facilitate T cell penetration into the center of tumors, and provoke antitumor immunity (*Mariathasan et al., 2018*). In the present study, we propose a model of autocrine signaling by cytokines in which VPS9D1-AS1 synergistically activates TGF-β and IFN signaling and induces cytokines in tumor cells to regulate CD8[+] T cell infiltration and differentiate into an exhausted phenotype.

AOM/DSS-induced murine colorectal cancers frequently carry mutations in *Ctnnb1* and *Kras* and chronic inflammation-related MSI (*Sharp et al., 2018*), the Wnt pathway is constitutively activated, closely mirroring human CRC at the molecular level (*Schulz-Heddergott et al., 2018*). The *Tg* model we studied here highlights the driving role of VPS9D1-AS1, which might have been activated by mutations caused by AOM/DSS. Chronic inflammation contributes to intestinal symptoms, such as bleeding (*Zhang et al., 2020*). ASOs are chemically synthesized single-stranded oligonucleotides that selectively inhibit the target mRNA, and it is covalently linked to hepatocyte asialoglycoprotein receptor-binding N-acetylgalactosamine (GalNAc) to achieve cell-selective gene silencing (*Yu et al., 2020*). ASO drugs targeting VPS9D1-AS1 inhibited tumor-associated symptoms such as bleeding and prolonged survival time. Thus, this preclinical model demonstrates that suppressing VPS9D1-AS1 is an effective way to treat CRC.

In summary, we described the mechanism by which VPS9D1-AS1 promoted tumor immune evasion through the TGF-β signaling pathways and ISGs and provided compelling evidence that the VPS9D1-AS1/TGF-β/ISG axis might serve as a drug target to enhance the efficacy of ICB treatment against CRC. These findings expand our current mechanistic understanding of CRC progression and provide potential therapeutic approaches by targeting VPS9D1-AS1 to enhance immunotherapy in patients with CRC.

# Materials and methods

## Key resources table

| Reagent type (species) or resource | Designation | Source or reference | Identifiers | Additional information |
|---|---|---|---|---|
| Antibody | TGF-β (Rabbit polyclonal) | Cell Signaling Technology | Cat# 3711 s | mfIHC(1 : 800) WB(1:200) IF (1:100) |
| Antibody | TGFBR1 (Rabbit polyclonal) | Abcam | Cat# ab31013 | IF (1:1000) mfIHC(1:2000) WB(1:500) IHC(1:400) |
| Antibody | TGFBR2 (Mouse monoclonal) | Abcam | Cat# ab78419 | WB(1:500) |
| Antibody | SMAD1/5/9 (Rabbit polyclonal) | Abcam | Cat# ab66737 | mfIHC(1:1000) WB(1:400) |
| Antibody | pSMAD1/5/9 (Ser 463/Ser465) (Rabbit polyclonal) | Santa Cruz Biotechnology | Cat# sc-12353 | mfIHC(1 : 500) |
| Antibody | SMAD4 (Mouse monoclonal) | Cell Signaling Technology | Cat# 46535 | mfIHC(1 : 1000) |
| Antibody | SMAD2/3 (Rabbit polyclonal) | Santa Cruz Biotechnology | Cat# sc-8332 | mfIHC(1 : 400) |
| Antibody | pSMAD2/3 (Ser 423/425) (Rabbit polyclonal) | Santa Cruz Biotechnology | Cat# sc-11769 | mfIHC(1 : 1000) |
| Antibody | SMAD6 (Rabbit polyclonal) | Santa Cruz Biotechnology | Cat# sc-25321 | WB(1:200) |
| Antibody | Cytokeratin (CK) (Mouse monoclonal) | Santa Cruz Biotechnology | Cat# sc-57004 | mfIHC(1 : 200) WB(1:500) |
| Antibody | STAT1 (Rabbit polyclonal) | Cell Signaling Technology | Cat# 14994 | WB(1:500) |
| Antibody | pSTAT1(Tyr701) (Rabbit polyclonal) | Cell Signaling Technology | Cat# 9167 | WB(1:500) |

*Continued on next page*

*Continued*

| Reagent type (species) or resource | Designation | Source or reference | Identifiers | Additional information |
|---|---|---|---|---|
| Antibody | ERK (Rabbit polyclonal) | Cell Signaling Technology | Cat# 4695 S | WB(1:200) |
| Antibody | pERK (Rabbit polyclonal) | Cell Signaling Technology | Cat# 4376 S | WB(1:200) |
| Antibody | Vimentin (Rabbit monoclonal) | Abcam | Cat# ab92547 | WB(1:100) |
| Antibody | E-cadherin (Rabbit polyclonal) | Abcam | Cat# ab15148 | WB(1:100) |
| Antibody | N-cadherin (Rabbit polyclonal) | Abcam | Cat# ab12221 | WB(1:100) |
| Antibody | GAPDH (Mouse monoclonal) | Beyotime | Cat# AG019-1 | WB(1:1000) |
| Antibody | β-actin (Mouse monoclonal) | Beyotime | Cat# AF0003 | WB(1:1000) |
| Antibody | CD4 (Rabbit monoclonal) | Abcam | Cat# ab133616 | IHC(1:200) mfIHC(1:1000) |
| Antibody | CD8 (Mouse monoclonal) | Santa Cruz Biotechnology | Cat# sc-53212 | IHC(1:500) mfIHC(1:200) |
| Antibody | FOXP3 (Rabbit polyclonal) | Cell Signaling Technology | Cat# 98377 | IHC(1:200) mfIHC(1:800) |
| Antibody | OAS1 (Rabbit polyclonal) | Cell Signaling Technology | Cat# 14498 | IHC(1:100) |
| Antibody | IFI27 (Rabbit polyclonal) | Abclonal | Cat# A14174 | WB(1:100) |
| Antibody | RPS3 (Rabbit monoclonal) | Abcam | Cat# ab128995 | WB(1:200) |
| Antibody | IFNAR1 (Rabbit polyclonal) | Abcam | Cat# ab45172 | WB(1:100) |
| Antibody | PDL1 (Rabbit monoclonal) | Abcam | Cat# ab205921 | WB (1:100) |
| Antibody | Ki67 (Rabbit polyclonal) | ZSGB-Bio | Cat# ZM-0166 | IHC (1:1000) |
| Antibody | Human CD3 (OKT3) FG (Mouse monoclonal) | eBioscience | Cat# 16-0037-85 | (1:100) |
| Antibody | Human CD28 FG 16-0289-85 (Mouse monoclonal) | eBioscience | Cat# 16-0289-85 | |
| Antibody | Human FITC CD3 (Mouse monoclonal) | BD Pharmingen | Cat# 555332 | FCM(1:100) |
| Antibody | Human APC CD4 (Mouse monoclonal) | BD Pharmingen | Cat# 555349 | FCM(1:100) |
| Antibody | Human PE CD8 (Mouse monoclonal) | BD Pharmingen | Cat# 555637 | FCM(1:100) |
| Antibody | Ultra-LEAF purified anti-mouse CD3 Antibody (Rabbit monoclonal) | BioLegend | Cat# 100239 | FCM (1:100) |
| Antibody | Ultra-LEAF purified anti-mouse CD28 Antibody (Mouse monoclonal) | BioLegend | Cat# 122021 | (1:100) |
| Antibody | Mouse FITC CD3 (Rabbit monoclonal) | BioLegend | Cat# 100204 | FCM(1:100) |
| Antibody | Mouse PE CD8a (Rabbit monoclonal) | BioLegend | Cat# 100708 | FCM(1:100) |
| Antibody | Mouse APC CD44 (Rabbit monoclonal) | BioLegend | Cat# 103012 | FCM(1:100) |
| Antibody | Mouse BV421 CD62L (Rabbit monoclonal) | BioLegend | Cat# 104435 | FCM(1:100) |
| Antibody | Mouse APC CD279 (PD-1) (Rabbit monoclonal) | BioLegend | Cat# 135209 | FCM(1:100) |
| Antibody | Mouse PE/Cy7 CD107a(LAMP-1) (Rabbit monoclonal) | BioLegend | Cat# 121619 | FCM(1:100) |
| Antibody | Mouse PE/Cy7 CD197 (CCR7) (Rabbit monoclonal) | BioLegend | Cat# 353211 | FCM(1:100) |
| Antibody | InVivoPlus Anti-mouse PD-1 (Mouse monoclonal) | BiXcell | Cat# BP0273 | (1:100) |
| Antibody | InVivoPlus Anti-mouse/human TGF-β (Mouse monoclonal) | BiXcell | Cat# BP0057 | (1:100) |

*Continued*

| Reagent type (species) or resource | Designation | Source or reference | Identifiers | Additional information |
|---|---|---|---|---|
| Antibody | Human IFN-alpha/beta R1 PE-conjugated antibody (Mouse monoclonal) | R&D | Cat# FAB245P | FCM(1:100) |
| Biological samples (*Homo sapiens*) | Human colon cancer tissue microarray | OUTDO | HCol-AS-180Su-14 | |
| Biological samples (*H. sapiens*) | Human rectal cancer tissue microarray | OUTDO | HRec-Ade180Sur-05 | |
| Biological samples (*H. sapiens*) | Human colorectal cancer tissue cDNA | OUTDO | HcolA30CS01 | |
| Cell lines (*H. sapiens*) | HCT116 | Dr. L. Yang (Beijing Chaoyang Hospital) | | |
| Cell lines (*H. sapiens*) | SW480 | Dr. L. Yang (Beijing Chaoyang Hospital) | | |
| Cell lines (*H. sapiens*) | RKO | Dr. L. Yang (Beijing Chaoyang Hospital) | | |
| Cell lines (*H. sapiens*) | HEK293T | Dr. L. Yang (Beijing Chaoyang Hospital) | | |
| cell lines (*Mus musculus*) | MC38 | PUMC | | |
| Cell lines (*M. musculus*) | CT26W | PUMC | | |
| Commercial assay or kit | QuantiNova SYBR Green PCR kit | Qiagen | Cat# 208054 | |
| Commercial assay or kit | QuantiTect Reverse Transcription kit | Qiagen | Cat# 205314 | |
| Commercial assay or kit | Poerce BCA Protein Assay Kit | Thermo | Cat# RB230514 | |
| Chemical compound, drug | Lipofectamine 3000 | Invitrogen | Cat# L3000015 | |
| Recombinant proteins | Human Interleukin-2 (hIL-2) | Cell Signaling Technology | Cat# 8907SC | |
| Commercial assay or kit | PerkinElmer Opal 7-color fIHC Kit | PerkinElmer | Cat# NEL797001KT | |
| Commercial assay or kit | Ficoll reagent | Solarbio | Cat# P8900 | |
| Commercial assay or kit | Tumor Dissociation Kit, mouse | Miltenyi Biotec | Cat# 130-091-376 | |
| Commercial assay or kit | CD8a$^+$ T Cell Isolation Kit, mouse | Miltenyi Biotec | Cat# 130-104-075 | |
| Commercial assay or kit | Anti-Human CD8 Magnetic Particles-DM | BD Pharmingen | Cat# 557766 | |
| Commercial assay or kit | HiScribeTM T7 Quick High Yeild RNA Synthesis Kit | NEB | Cat# E2040S | |
| Commercial assay or kit | Monarch RNA Cleanup Kit | NEB | Cat# T2030S | |
| Commercial assay or kit | IP Lysis Buffer | Thermo Fisher | Cat# 87787 | |
| Commercial assay or kit | Pierce RNA 3' Desthiobiotinylation kit | Thermo Fisher | Cat# 20163 | |
| Commercial assay or kit | PIERCE MAGNETIC RNA PULL-DOWN KIT | Thermo Fisher | Cat# 20164 | |
| Chemical compound, drug | CX-5461 | MCE | Cat# HY-13323 | |
| Recombinant proteins | Human Interferon-α1 | Cell Signaling Technology | Cat# 8927SC | |
| Commercial assay or kit | DIG Northern Starter | Roche | Cat# 12039672910 | |
| Commercial assay or kit | VPS9D1-AS1 FISH Probes | Biosearch | N/A | |
| Commercial assay or kit | Magna RIP RNA-Binding Protein Immunoprecipitation Kit | Millipore | Cat# 17–700 | |
| Commercial assay or kit | Positively Charged Biodyne Nylon membrane | Invitrogen | Cat# 10100 | |

*Continued on next page*

*Continued*

| Reagent type (species) or resource | Designation | Source or reference | Identifiers | Additional information |
|---|---|---|---|---|
| Commercial assay or kit | ChIP-IT Express Enzymatic Magnetic Chromatin Immunoprecipitation kit | Active Motif | Cat# 53009 | |
| Commercial assay or kit | Stellaris RNA FISH Wash Buffer A | BIOSEARCH | Cat# SMF-WA1-60 | |
| Commercial assay or kit | Stellaris RNA FISH Wash Buffer B | BIOSEARCH | Cat# SMF-WA1-20 | |
| Commercial assay or kit | Stellaris RNA FISH Hybridization buffers | BIOSEARCH | Cat# SMF-HB1-10 | |
| Commercial assay or kit | Stellaris FISH Probes with Quasar 570 Dye | BIOSEARCH | Cat# SMF-1063–5 | |
| Commercial assay or kit | FITC labeled Goat anti-rabbit secondly antibody | Byotime | Cat# 0652 | |
| Commercial assay or kit | FITC labeled Goat anti-mouse secondly antibody | Byotime | Cat# A0568 | |
| Commercial assay or kit | Antifade mounting medium with DAPI | Solarbio | Cat# S2110 | |

## RNAscope in situ hybridization assay

RNA in situ hybridization assays were performed using an RNAscope kit (Advanced Cell Diagnostics, Hayward, CA, USA). VPS9D1-AS1 mRNA molecules were detected with single-copy detection sensitivity. Single-molecule signals were quantified on a cell-by-cell basis by manual counting. The signals per cell were evaluated to quantitate the levels of VPS9D1-AS1. The signals were graded as 0 (0–1 dots/10 cells), + (1–3 dots/cell), ++ (4–10 dots/cell), +++ (>10 dots/cell with <10% of dots in clusters), and ++++ (>10 dots/cell with >10% of dots in clusters) (*Zhao et al., 2018*). The scores were classified as follows: '−, +' represented negative expression and '++, +++, ++++' represented positive expression of VPS9D1-AS1.

## Mice

Conditional floxed human VPS9D1-AS1 knock-in alleles with LoxP sites were introduced into the *Rosa26* gene locus to construct C57BL/6J-Gt (ROSA)26Sor^em(CAG-VPS9D1-AS1)1Smoc mice. A mixture of the construct and CRISPR/Cas9 vector was microinjected into zygotes. The zygotes were implanted into foster mice. The above procedures were performed by Shanghai Model Organisms. Successful integration in the founder mice was identified by PCR analyses of genomic DNA using primers targeting VPS9D1-AS1. After screening, the positive founder (bearing R26-eCAG-VPS9D1-AS1) was crossed with Cre-*Villin* mice to produce conditional VPS9D1-AS1 transgenic (VPS $^{Tg}$) mice in the intestinal epithelium. VSP$^{Tg}$ mice were backcrossed to C57BL/6 mice for at least six generations. The VPS $^{Tg}$ mice did not show any differences in comparing with WT mice in lifespan and body weight.

To induce colorectal cancer with AOM/DSS, C57BL/6 VPS $^{Tg}$ mice received a single intraperitoneal injection of AOM (Sigma Aldrich) at a dose of 10 mg/kg body weight. One week later, animals were exposed to 1–3 cycles of 2% DSS (MW = 36,000–50,000, MP Biomedicals, CAT# 216011050). For in vivo inhibition of VPS9D1-AS1, mice in the treatment group received 10 nmol (100 µl, 100 mM ASO in PBS) ASOs i.p., once a week. Intestinal tumor volumes were calculated by the formula: $\pi \times (1/2$ width$)^2 \times$ length.

For the xenograft tumor model, $2 \times 10^6$ HCT116 or SW480 cells were implanted subcutaneously in 6-week-old female BALB/c nude mice (Charles River). For MC38 cells, $4 \times 10^6$ cells were subcutaneously injected into the lower abdominal region of 6-week-old C57BL/6 WT mice (Charles River). Tumors were assessed two to three times a week by a caliper measurement, and the volumes were calculated with the following formula: $1/2 \times$ (length $\times$ width$^2$). MC38 cells were stably transfected with luciferase to label tumor progression and were monitored using a Carestream in vivo imaging system (MS FX Pro). Animal experimental protocols were approved (AEEI-2021–105) according to the guidelines of the Ethics Committee for Animal Testing of Capital Medical University.

## Patient samples

There were two independent cohorts in this study. OUTDO cohort enrolled the colon and rectal cancer tissue microarrays (TMA) and cDNA sample array, which were provided by Shanghai OUTDO

Biotech (Shanghai OUTDO Biotech Co., Shanghai, China). The HCol-Ade 180Sur-07 TMA consisted of tumor tissues and matched normal tissues from 90 patients with colon cancer who underwent surgery from January 2008 to November 2009 with follow-up information available until May 2014. The Hrec-Ade180Sur-04 TMA included tumor tissues and matched normal tissues from 90 patients with rectal cancer who underwent surgery from October 2007 to August 2008 with follow-up information available until September 2014. Colon cancer cDNA sample array enrolled 80 colon patients and consisted with 80 cDNA samples from tumor tissues and 15 cDNA samples from normal tissues. Bei Jing Chao Yang Hospital (BJCYH) cohort enrolled the tissue and blood samples to use for validation. Tumor samples were fresh or formalin-embedded and collected from CRC patients from 2020 to 2021 at Beijing Chao-yang Hospital.

All sample donors provided informed consent, and the study was conducted under the approval of the Institutional Ethics Committee from Beijing Chaoyang Hospital of Capital Medical University between 2018 and 2020 (2018-ke-24). Samples were collected from patients with CRC who did not receive chemotherapy or radiotherapy before surgery. All procedures were performed in accordance with the relevant guidelines and regulations.

## RNA fluorescence in situ hybridization (FISH) and immunofluorescence (IF)

Cells were seeded in 24-well plate with glass coverslips (Solarbio), fixed with 4% (w/v) paraformaldehyde (Solarbio) for 10 min at room temperature, washed with 1× PBS, 70% ethanol for 1 hr, and washed with Stellaris Buffer A (Biosearch). Then cells were incubated with Stellaris FISH probes which were diluted by Stellaris FISH Hybridization Buffer (1:100) at 37°C overnight. Cells were next washed with Buffer A for 30 min at 37°C and B for 5 min at room temperature. Primary antibodies TGF-β (dilution, 1:100), TGFBR1 (dilution, 1:1000), and SMAD1/5/9 (dilution, 1:500) with 5% BSA in PBS were, respectively, incubated with cells for 2 hr at room temperature. Then, cells washed three times with PBS and incubated with fluorescein isothiocyanate (FITC)-labeled goat anti-rabbit/mouse secondary antibody (Beyotime) for 30 min at room temperature. Finally, coverslips were mounted onto slides with antifade mounting medium supplemented with DAPI (Solarbio). Confocal images were acquired using a Lecia TCS SP8 laser scanning microscope.

## CRISPR/Cas9 interference

CRISPR/Cas9 (all-in-one lentiviral) system was employed to achieve KO of target genes. Briefly, the predicted gRNAs were designed by online tools (http://crispr.mit.edu/) and individually cloned into the all-in-one lentiviral vector (LnetiCRISPRv2). To obtained KO lentiviruses, lentiviral vector along with the packing plasmids psPAX2 and pMD2.G (VSV-G envelope) was transfected into HEK293T cells using Lipofectamine 3000 (Invitrogen) in serum/antibiotic-free media. Media were collected at second and third days post-transfection. The lentiviral media were used to transduce target cells after filtering through a 0.45-µM filter to remove cells or debris and supplied 10 µg/ml Polybrene (Santa Cruz Biotechnology). To produce stable KO models, cells were selected with puromycin for at least 14 days. The KO efficiencies were confirmed by qRT-PCR, northern blotting, or western blotting. The sgRNAs sequences were listed in *Supplementary file 1a*.

## RNA immunoprecipitation

Tumor cells were harvested at 80% confluence that were seeded in a 10-cm dish for RIP. RIP kit (Magna RIP RNA-Binding Protein Immunoprecipitation Kit, Millipore) was applied to detect the interactions between VPS9D1-AS1 and intended antibody-protein complex. The procedures were carried out according to the manufacturer's instructions. Mouse or Rabbit normal IgG (Millipore), which was served as negative control. InPut and co-immunoprecipitated RNAs were extracted by the reagents in RIP kit for further analysis by qRT-PCR.

## Northern blot

For synthesize probes that targeted VPS9D1-AS1, PCR reaction was conducted to clone a 722 bp sequence from pcDNA3.1-VPS9D1-AS1 templated. Primers were listed in *Supplementary file 1b*. RNA probes were transcript by T7 RNA polymerase and labeling by digoxigenin-11-UTP. To visualize VPS9D1-AS1, a total of 10 µg of the indicated RNA from HCT116 or SW480 cells were subjected to

formaldehyde gel electrophoresis and transferred to a positively charged Biodyne Nylon membrane. After fixation, the membranes were hybridized with VPS9D1-AS1 probes at 68°C for 12 hr. At last, anti-digoxigenin-AP was incubated with the membrane for 30 min at room temperature. Washes were performed as described in DIG Northern Starter Kit, and the membrane was placed in CDP-Star solution and visualized by a Bio-Rad ChemiDoc XRS + system.

## RNA isolation, cDNA synthesis, qRT-PCR

Total RNA was extracted from cell or tissue samples using TRIzol reagent (Invitrogen, Carlsbad, CA, USA). For determining mRNA, total RNA (1 µg) was reverse transcribed using the QuantiTect Reverse Transcription kit (Qiagen). qRT-PCR was performed on cDNA using the Applied Biosystems 7500 Real-Time PCR System (Life Technologies, Gaithersburg, MD, USA) and the SYBR Select Master Real-Time PCR assay (Qiagen). For determining microRNA, reverse transcriptions were performed by miScript II RT KIT (Qiagen), and qRT-PCR reactions were carried out by miScript SYBR Green PCR Kit (Qiagen). The PCR program was as follows: predenaturation at 95°C for 5 min, 40 cycles of denaturation at 95°C for 5 s, and annealing 60°C for 30 s and elongation at 72°C for 1 min. The primers are listed in *Supplementary file 1b*.

## Small interfering RNA (siRNA), microRNA mimic transfection

siRNAs or microRNA mimics used to silence targets are listed in *Supplementary file 1c*. Indicated siRNAs or microRNA mimics were transient transfected into targeted cells by Lipofectamine 3000.

## Western blotting

Cells were lysate and determined the protein concentrations using BCA method (Thermo). Then, these lysates were prepared to protein solution and separated by sodium dodecyl sulfate polyacrylamide gel electrophoresis (SDS-PAGE). SDS-PAGE gel was subjected to semi-dry transfer and transferred proteins to a polyvinylidene fluoride membranes (Millipore, MA, USA). Follow blocking using 5% milk, membrane was incubated with various primary antibodies for overnight at 4°C. After washing, goat anti-mouse/rabbit IgG secondary antibodies (LI-COR) were incubated the membranes for 1 hr at room temperature. Membrane was visualized using an LI-COR odyssey imager. The relative levels of intended proteins were calculated using ImageJ software (Version 1.52 a).

## CCK8, transwell migration, and soft agar clone forming assays

For cellular proliferative ability assays, 800 tumor cells seeded in 96-well plates and incubated with 100-µL culture medium for 5 days. Cellular viability was determined by Cell Counting Kit (CCK8, Dojindo) with dilution 1:100 at each day. For cell migration assays, $5×10^4$ cells were seeded in the upper transwell chambers (8 µm pore size; Coring) using the medium with fetal bovine serum while supplied complete medium in 24-well plate. After culturing for 48 hr, medium and non-migrated cells in upper chambers were wiped off with cotton swaps, and the filter was stained for 30 min with 0.04% crystal violet and rinsed with PBS for three times. For cell clone forming assays, $2×10^4$ tumor cells were seeded in 12-well plates that were coated with dulbecco's modified eagle medium (DMED) containing 1.2% (w/v) soft agar and culture for 30 days. Colonies were stained using 0.04% crystal violet and counted.

## mRNA expression profile sequencing

All sequencings were performed by Novogene. Total RNA was extracted from three HCT116[sgVPS] and three HCT116[sgControl] cell samples. The quality of purified RNA was tested on an Agilent 2100 Bioanalyzer system (Agilent technologies, Santa Clara, CA, USA). The ribosomal RNAs were removed and were fragmented into small pieces. The first strands were reverse-transcribed into first cDNA by random primer, RNAs were digested by RNase H, followed by second strand cDNA synthesis using DNA polymerase I. A single 'A' base adapter was added to the fragments. AMPure XP beads were used filtered 370–420 bp small pieces. The products were removed by second strands with U base using USER enzyme and amplified by PCR to create the final cDNA library. The raw reads were cleaned and mapped against the human reference genome using Hisat2. The differential level of genes was determined based on the value of Fragments Per Kilobase of transcript sequence per Millions

base pairs sequenced, which was calculated by cuffdiff. GSEA was performed to enrich pathway of VPS9D1-AS1 based on the differential genes using RNA sequencing.

## Immune subtype and consensus molecular subtype (CMS) analyses using TCGA

RNAseq data for TCGA COAD and READ tumor tissues were downloaded from Broad GDAC Firehose portal (https://gdac.broadinstitute.org/). The transcripts per kilobase of exon model per million mapped reads (TPM) values were used to validate the associations for immune subtypes and CMS of indicated genes. The status of immune subtype for each TCGA sample was recognized according to the criterion from *Thorsson et al., 2018*. CMS information was derived from http://www.synapse.org (*Guinney et al., 2015*).

## RNA pulldown assay

Four probes (RNA probes for RPD) that targeted four regions of VPS9D1-AS1 transcript were designed to perform RPD assay. The primers were listed in *Supplementary file 1d*. RNA fragments were transcribed with HiScribeTM T7 Quick High Yeild RNA Synthesis Kit (NEB) in vitro. RNA transcript products were treated with RNAase-free DNase I on column with Monarch RNA Cleanup Kit (NEB). These RNAs were labeled using Pierce RNA 3' Desthiobiotinylation kit (Thermo). $1 \times 10^7$ cell samples were lysed by IP Lysis Buffer (Thermo). Pulldown assays were performed following the Pierce Magnetic RNA-Protein Pull-Down Kit (Thermo). Briefly, RNA probes were bound to streptavidin magnetic beads. Then, these beads were mixed with cell extract and incubated at 4°C for 60 min with rotation. The binding protein complexes were heated at 95–100°C for 10 min for purifying intended proteins and further analyzed by western blotting.

## Chromatin immunoprecipitation (ChIP) and quantitative PCR

A ChIP assay for various nuclear proteins was performed using ChIP-IT Express Enzymatic Magnetic Chromatin Immunoprecipitation kit (Active Motif). In brief, $1 \times 10^7$ cells were fixed with fixation solution at room temperature for 10 min and stopped by adding glycine buffer. The cells were resuspended in ice-cold lysis buffers provided in the kit and homogenized to aid in nuclei release. To sheared DNA (200–400 bp), nuclei solutions were resuspended and digested for 5 min at 37°C by enzyme buffers provided in the kit. For immunoprecipitation reactions, samples were incubated with 1 µg of intended antibodies or isotype control (rabbit/mouse IgG) and Protein G Magnetic Beads for overnight at 4°C. The magnetic beads were washed by buffers in the kit. The DNA-protein complex was eluted by heating at 95°C for 15 min in a thermocycler. The DNA were isolated by adding proteinase K to digest binding proteins and stopped by stopping buffer in kit. The DNA was then subjected to real-time PCR or routine PCR analysis. The primers for ChIP-PCR were listed in *Supplementary file 1e*.

## Multispectral fluorescence immunohistochemistry (mfIHC) and scoring multispectral images

For quantitatively determining the infiltrated lymphocytes, multispectral imaging was employed to stain antibodies in a TMA slide. The mfIHC assays were performed using PerkinElmer Opal 7-color fIHC Kit (PerkinElmer) according to manufacturer's introduction. Slide was baked 2 hr at 65°C to prevent tissue samples fall out from slide. After that, slide was deparaffinized to water following the cycle, xylene 15 min two times, 100% ethanol, 95% ethanol, 85% ethanol, and 75% ethanol. Then, samples were antigen retrieval using microwaving method (90 s on 100% power, followed by 15 min on 20% power). The primary antibodies against CD4 (Abcam, dilution 1:1000), CD8 (Santa Cruz, dilution 1:200), FOXP3 (CST, 1:800), and CK (Santa Cruz, dilution 1:200), respectively, stained for 1 hr at room temperature and then incubated with horseradish peroxidase-labeled mouse/rabbit secondary antibody for 10 min at room temperature. Tyramide (TSA)-conjugated fluorophore, a series Opal reagent, mocked the slide at 1:100 dilution. Opal 690 labeled CD4, Opal 570 labeled CD8, Opal 620 labeled FOXP3, and Opal 520 labeled CK. Slide was heated using microwave to remove primary antibody and leaves Opal reagent to show target protein at the last of each Opal staining procedure. Finally, slide was counterstained with Spectral DAPI to show nuclei and cover slips followed by sealing with mounting medium supplied with antifade solution (Applygen Technologies Inc Beijing, China).

Multispectral TMA images were scanned using Vectra Polaris Imaging System (Perkin Elmer) through ×10 objective lens.

InForm 2.1.1 software (Perkin Elmer) was used to batch analysis of multispectral images. The scanned image of each tissue core was loaded to InForm and substrate background using the image derived from a slide without staining in the same exposure setting. The segmented tissues of parenchymal neoplasm and mesenchyme stroma were trained according to CK signals intensity. Cells were segmented to count the total number of cells. The percentages for CD4, CD8, and FOXP3 T cells were calculated using the number for each stained by Opal signal dividing by total cell number in tumor or stroma, respectively.

## Immunohistochemistry (IHC) assays

The primary antibodies used in immunohistochemistry (IHC) for tissue slides were used for IHC of tissue slides: anti-IFNAR1 (Abclonal, 1:200 dilution), anti-CD4 (Abcam, 1:200 dilution), anti-CD8 (Abcam, Cat# ab93278, 1:500 dilution), anti-FOXP3 (CST, 1:200 dilution), anti-OAS1 (CST, 1:100 dilution), and anti-TGFBR1 (Abcam, 1:400 dilution). The slices were deparaffinized and rehydrated, pretreated with a citric acid antigen retrieval solution (pH = 6.8), and rinsed in PBS. The sections were blocked in 2% goat serum and incubated with the primary antibody overnight at 4°C. The streptavidin-peroxidase method (ZSGG-BIO, CAT# PV9001) was used to show the levels of stained proteins. The number of CD4, CD8, and FOXP3 cells were calculated using K-Viewer V1 system (Version, Code 1.5.3.1, KONFOONG BIOTECH INTERNATIONAL CO., LTD). The levels of OAS1, IFNAR1, and TGFBR1 were scored as follows: 1 (0–25%), 2 (26–50%), 3 (51–75%), and 4 (>75%). The intensity of positive staining was classified into four scales as follows: 0 (negative), 1 (weak), 2 (moderate), and 3 (strong). The levels were semiquantitatively determined as percentages multiplied by intensity.

## CD8$^+$ T cell sorting and culturing

Anti-human CD8 magnetic particles (BD) were used to positive selection of human CD8-bearing leukocytes. Human peripheral blood mononuclear cells (PBMC) were isolated by density gradient centrifugation using Ficoll reagent (Solarbio). 50-µl CD8 magnetic particles were mixed with per $1×10^7$ PBMCs and incubated at room temperature for 30 min. The particle labeling volume was placed on the cell separation magnet and washed with buffers in kit. The positive fractions were resuspended with RPMI1640 media. CD3 antibodies, CD28 antibodies, and hIL2 were added to media to expand CD8$^+$ T cells.

Splenocytes were harvested from the spleen of OT-I mice using Mouse CD8$^+$ T Cell Isolation Kit (Miltenyi). In brief, 10-µl Biotin-Antibody Cocktail were mixed with per $10^7$ cells and incubated for 5 min at 4°C. Then, 20-µl Anti-Biotin MicroBeads were added and incubated for 10 min at 4°C. Above cell suspensions were subsequent applied to LS Column in the magnetic field of Magnetic Activated Cell Sorting (MACS) Separator (Miltenyi). The flow-through that contained unlabeled CD8$^+$ T cells was collected and cultured with mouse CD3 and CD28 antibodies for an additional 3 days before use in co-culture.

## In vitro cytotoxicity assays

$5×10^5$ tumor cells were seeded per well in 12-well culture plate (Excell Bio). At first, CD8$^+$ T cells were admixed in dilutions (twofold, starting at a 1:1 ratio) to determine the cytotoxic efficiencies. After co-culture for 24 hr, T cells were washed away and collected for FCM sorting. After further 3 days, plates were fixed and stained for 30 min using a 0.04% crystal violet solution (Solarbio). When indicated, blocking TGF-β and PD-1 antibodies (BioXcell) was added in media (5 µl per well).

## Flow cytometry

For in vitro cell lines, $5×10^5$ cells were digested by 0.25% pancreatin enzymes (Biological Industry) to single-cell suspensions. Cell surface IFNAR1 staining was done for 1 hr at room temperature. Intracellular IFNAR1 staining was fixed using 4% (w/v) paraformaldehyde (Solarbio) and permeabilized using 0.1% (v/v) Trxion-100 (Amresco). For infiltrated T cells in xenograft tumors, single-cell suspensions were prepared by mouse Tumor Dissociation Kit (Miltenyi) according to the manufacturer's instructions. Mouse spleens were cut into pieces and filtered using a Coring cell strainer to prepare single-splenocyte suspensions. Both splenocyte and PBMC suspensions were lysed using 1× lysis buffer (BD)

to remove red cells. A BD FACScanto II was used for data acquisition, and analysis was performed using FlowJo (TreeStar).

## Statistical analysis

Statistical analysis was performed using GraphPad Prism software and R software. Student's *t* test or ANOVA (one- or two-way) with *Bonferroni* post hoc test was used to evaluate the statistical significance. The survival curves were plotted according to the Kaplan-Meier method and evaluated by the log-rank test. A p-value less than 0.05 was considered statistically significant.

## Acknowledgements

We thank Dr. Jian Liu for providing CRISPR/Cas9 virus packaging vectors and Dr. Suliang Guo for his assistance in animal feeding.

This project was supported by grants from the National Natural Science Foundation of China (81802349, 82173234), Beijing Natural Science Foundation (7192070), Beijing Municipal Administration of Hospitals Incubating Program (PX2018013), Scientific Research Project of Beijing Educational Committee (KM201910025016), and Open Project of Key Laboratory of Cardiovascular Disease Medical Engineering, Ministry of Education (2019XXG-KFKT-03).

## Additional information

### Funding

| Funder | Grant reference number | Author |
|---|---|---|
| Natural Science Foundation of China | 81802349 | Lei Yang |
| National Science Foundation of China | 8213234 | Tao Wen |
| Beijing Natural Science Foundation | 7192070 | Lei Yang |
| Beijing Municipal of Hospitals Incubating Program | PX2018013 | Lei Yang |
| Scientific Research Project of Beijing Educational Committee | KM20190025016 | Lei Yang |
| Open Project of Key Laboratory of Cardiovascular Disease Medical Engineering, Ministry of Education | 2019XXG-KFKT-03 | Lei Yang |

The funders had no role in study design, data collection and interpretation, or the decision to submit the work for publication.

### Author contributions

Lei Yang, Conceptualization, Data curation, Funding acquisition, Writing - original draft, Project administration, Writing - review and editing; Xichen Dong, Validation, Investigation, Methodology; Zheng Liu, Investigation, Methodology; Jinjing Tan, Visualization, Methodology; Xiaoxi Huang, Methodology; Tao Wen, Conceptualization, Funding acquisition, Visualization; Hao Qu, Zhenjun Wang, Conceptualization, Resources, Project administration

### Author ORCIDs

Lei Yang ⓘ http://orcid.org/0000-0003-3718-2138

## Ethics

Human subjects: All sample donors provided informed consent, and the study was conducted under the approval (2018-ke-24) of the Institutional Ethics Committee from Beijing Chaoyang Hospital of Capital Medical University between 2018 and 2020 samples were collected from patients with CRC who did not receive chemotherapy or radiotherapy before surgery.

Animal experimental protocols were approved (AEEI-2021-105) according to the guidelines of the Ethics Committee for Animal Testing of Capital Medical University.

## Decision letter and Author response

Decision letter https://doi.org/10.7554/eLife.79811.sa1
Author response https://doi.org/10.7554/eLife.79811.sa2

## Additional files

### Supplementary files

• Supplementary file 1. Sequences for sgRNA, PCR primer, siRNA and shRNA. (a) sgRNA sequence. (b) Primers for PCR. (c) siRNA and shRNA sequences. (d) The primer for RNA pull down (RPD) probe synthesized. (e) The Primers for ChIP-PCR.

• MDAR checklist

### Data availability

RNA sequencing data set of HCT116 sgControl and sgVPS cells were deposited in Sequence Read Archive (PRJNA716724) and Dryad Digital Repository (https://doi.org/10.5061/dryad.qnk98sfk6).

The following datasets were generated:

| Author(s) | Year | Dataset title | Dataset URL | Database and Identifier |
|---|---|---|---|---|
| Yang Lei, Dong X, Liu Z, Tan J, Huang X, Wen Tao, Qu Hao, Wang Z | 2021 | VPS9D1-AS1 regualtes differential mRNA in HCT116 cells | https://www.ncbi.nlm.nih.gov/bioproject/PRJNA716724 | NCBI BioProject, PRJNA716724 |
| Yang L, Dong X, Liu Z, Tan J, Huang X, Wen T, Qu H, Wang Z | 2022 | Overexpression of VPS9D1-AS1, an activator of transforming growth factor β signaling, upregulates interferon-stimulated-gene expression to regress CD8+ T cell infiltration in the microenvironment of colorectal cancer | https://dx.doi.org/10.5061/dryad.qnk98sfk6 | Dryad Digital Repository, 10.5061/dryad.qnk98sfk6 |

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
