## [Editor Report]

This research work uncovered the role of a long noncoding RNA VPS9D1-AS1(VPS) in mediating immune evasion of colorectal cancer cells, which is achieved via amplifying intra-tumoral TGF-β/ISG signaling to facilitate escape from cytotoxic T cells killing. Overall, the experiments were well-designed and the data were properly analyzed. The findings are of potential significance to gaining insight into treating colorectal cancer cells by enhancing the efficacy of immunotherapy

---

## [Decision Letter]

**Decision letter after peer review:**

Thank you for submitting your article "VPS9D1-AS1 overexpression amplifies intratumoral TGF-β signaling and promotes tumor cell escape from CD8^+^ T cell killing in colorectal cancer" for consideration by *eLife*. Your article has been reviewed by 3 peer reviewers, 0one of whom is a member of our Board of Reviewing Editors, and the evaluation has been overseen by W Kimryn Rathmell as the Senior Editor. The following individual involved in the review of your submission has agreed to reveal their identity: Philippe Krebs (Reviewer #2).

Essential revisions:

1) The mechanistic study regarding VPS9D1-AS1/TGF-β/ISG was mainly carried out in vitro, while this part shall be tested in cell line-derived xenografts or mouse tumors to support the claims.

2) The data volume is large and the methodology has covered a broad range in the work, while the arrangement of the manuscript could be improved to optimize the presentation and enhance the readability.

3) Some details of the figures conflict with the statement in the manuscript, and corrections are warranted.

*Reviewer #1 (Recommendations for the authors):*

Here are some major concerns:

Figure 1/S2: The authors shall explain or discuss why FOXP3 T cells showed a different trend in OUTDO and BJCYH cohorts.

Figure 2: Which cohort(s) was analyzed in 2F? Did OUTDO and BJCYH show the same or different patterns?

Figure 4/S6: 4H-I showed that at 0 hours, OAS1 and IFI27 did not change between CTRL and VPS OE; however, in S6H, OAS1 and IFI27 were much lower in VPS OE, conflicting with 4H-I.

Figure 5/S7: In S7A, VPS OE+sgOAS1 (last three lanes) show a substantial OAS1 expression. This made the claims on OAS1 (5B) less convincing. How did VPS KO cells impact IFNAR1 level of T cells in coculture?

Figures 6 and 7: The mechanistic model proposed by in vitro studies (Figures 4 and 5) shall be tested in cell line-derived xenografts or mouse tumors. For example, the IFNAR1 level of VPS OE tumors was examined (7H) but the IFNAR1 level of infiltrated T cells was not. Moreover, CRC patient-derived xenografts shall be used and treated with VPS ASO, which will be more physiologically relevant than the cell line with 600-fold VPS upregulation used in the study.

*Reviewer #2 (Recommendations for the authors):*

My comments are detailed below:

(Figure 1)

1. Panels A and B (and S1A) are not useful for the conclusions of the paper – the histology is difficult to distinguish and it is not clear of levels (of expression) or frequencies (of positive cells) were quantified in panel B. Those data are better presented in panel D.

2. Panel F, for instance, doesn't actually corroborate what is indicated in panel A, as the authors already gate on "high" populations, and there are still some samples with almost no VPS9D1-AS1, this suggests the staining protocol is not a good replacement for qRT PCR.

3. In the text the authors write that VPS expression doesn't correlate with the expression of genes in the TGFb pathway (pp4, lines 18-20), but do not actually separate by gene expression, but rather by cancer and normal tissue. A better analysis may actually be a xy graph with regression curves.

(Figure 2)

1. There seem to be artefacts or resolution issues in panel A; in panel B, the inlays seem to have different magnifications (size bars are not provided).

2. It seems data were mixed up data in panel E; in addition, panel D shows Ca/STM ratios while panel E indicates cell numbers, making it difficult to compare the two cohorts. The same applies to Sfig2 panel C versus F.

3. Panel F needs to be more clearly explained, it is difficult to interpret these data with the explanation given and certain p values are missing. The same applies to SFigure panel G.

4. Panel A doesn't show what is indicated in the text (cancer vs stromal pp 5 lines 2-3). It shows neg. vs pos. according to the figure and it is difficult to distinguish healthy from tumor tissue.

(Figure 3)

1. Is SMAD1/5/9 in Figure 3D indeed the top band?

2. In panels E and M (and several other western blot data), automated quantification would help support the statements by the authors on increased / lower expression of specific proteins, which are not always convincing as stated.

3. L – the oligonucleotide design is not fully clear to me; how can these differences between probes be justified, e.g. between RPD1 and RPD2? Why not compare to the entire VPS9D1-AS1 molecule?

4. O – these data are not convincing.

Figure 4

1. In panel E it seems the labels have been swapped.

2. In panel F: there are differences in the two cell lines after VPS-KO; are the labels correct?

3. Panel J and references indicated in the text: I could not find in the literature evidence for a nuclear function of TGFb.

4. In panel K: how were the decimal values for OAS1, IFNAR1, and TGFBR1 determined?

(Figure 5)

1. Panel B: some of the samples seem to have been swapped. Please cross-check.

2. Panel D: How were patients selected to be either VPS neg. or pos.? This is also relevant due to the apparent intra-tumor heterogeneity of VPS9D1-AS1 expression. FACS plot, VPS positive condition, upper and lower rows: both T cell and non T cell populations are shifted down, respectively up, suggesting a problem or changes in the settings of the flow cytometer instrument. The quality of the flow cytometry data in panel G is also not convincing (several cells are out of the depicted area). Panel E and F: the authors should provide information on the cell types that have been gated on.

3. One concern about the T cell killing (panel F) is that the differences could be due to cell growth changes in the VPS OE cells as seen previously. Is there a way to account for/eliminate that in this setting? Check the units on the y axis of the graph. There is no clear effect of neutralizing antibodies (the conditions including MC38 cells + OT-1 cells should be compared).

(Figure 6).

1. The murine counterpart of VPS9D1-AS1 is mentioned in the text, but all experiences in vivo are performed using (human) VPS9D1-AS1. The rationale for not using NR045849 in wild-type mice should be better justified.

2. Panel B: are there differences in vitro in the growth of control and VPS OE MC38 cells? This should be indicated to better understand the in vivo phenotype of these cells.

3. In general, the flow cytometry data are not convincing. Some cell populations are squished on the top of the graph, there is no indication of what cells are gated on (no gating path or gating strategy indicated). Have isotype controls (or FMO) been used for PD1? The label of the X axis in E should be revised (2 markers on the same axis). If Figure 6D tumor left panel: CD8 numbers are close to 0; therefore, I am not sure how confident one can be about %PD1 in Figure 6F.

(Figure 7)

1. There should be an indication of the phenotype of VPS9D1-AS1 transgenic mice at steady-state; are these animals normal? And is VPS9D1-AS1 indeed only expressed in epithelial cells (this is not clear from FigS9B).

2. "However, Ifnar1 levels in tumor tissues were increased in VPS Tg mice (Figure 7H), indicating that the tumor-promoting effect of VPS9D1-AS1 OE was dependent on Ifnar1 expression": this is a strong statement or a conclusion that is not supported by the current data. Only a blockade of IFNAR (pharmacologically or genetically) would actually prove this statement.

3. What are the units on the Y axis for G? and how were the cells quantified?

4. Panel H: there seem to be differences in the quality of these stainings. It would be helpful to show healthy intestinal tissue (note that the current magnification does not allow to discern well the malignant tissue). How were IFNARI levels quantified? Note that the same comments also apply to panel N.

5. I think the author over-interpret the panel K data; again (semi-)quantitative densitometry would help to support their statements.

The authors do not discuss how the T cells and the VPS9D1-AS1 expressing tumors interact in vivo. Is this also mediated via TGFb (note that this is not shown)? And how do they explain that IFNARI becomes increased on T cells in the tumor micro-environment?

As mentioned in the public review section, the paper could be better served in several ways:

(1) The paper could be split into two parts. The paper has plenty of data to support the hypothesis of VSP importance in cancer with a less exhaustive look at the binding partners/etc. In particular, the data on downstream miRNAs (Figure 3O, currently not convincing) would benefit from a more in-depth analysis, possibly in a follow-up study that may as well include some of the data presented in Figure 4.

(2) The data could be slimmed.

A few additional suggestions:

– Introduction: the already reported roles of VPS9D1-AS1 for cancer in general and colon cancer, in particular, could be presented in greater detail.

– Figure 3A: For non-specialists, it would be helpful to indicate which cells originate from CRC tissue.

– Figure 3: why swap the colors from F and G? It's confusing.

– Figure 3: K – the units seem to be wrong in the y axis.

– Sfig4G: data on SW480 are too faint to be assessed.

– Fig6 and Fig7: these data are not from xenograft experiments since murine cells were injected into mouse recipients (of the same haplotype).

– Fig7 panel C (and several of the histology shown): larger magnification should be provided and grading by a pathologist would be beneficial.

– Overall, the data are very dense as presented. Their presentation could be improved to enhance readability.

*Reviewer #3 (Recommendations for the authors):*

1. Figures to present the expression difference of VPS9D1-AS1 between benign and malignant tissue should be consistent in Figure 1D and 1E for better visualization;

2. The P value between VPS 0 and VPS1 was more prominent than that between VPS 0 and VPS 2-4 in Figure 1F, while the Figure 1G only compares the difference between stroma and VPS 4, please explain the reason;

3. In Figure 1I, the difference in the expression level of TGFBR1, SMAD1/9, VPS9D1-AS1, and TGF-β seems not significant between the normal and cancer tissue, which is not consistent with the description in the manuscript;

4. The quantification of immune cell subsets was presented with two staining methods in Figure 2A and 2B, please specify the reason;

5. The number of CD8^+^ T cells in VPS9D1-AS1 positive tissue overweighs that in VPS9D1-AS1 negative tissue, which is contrary to the description in the manuscript;

6. Tumor pictures would be better provided along with the Figure 3G, as was done in Figure 7I;

7. On page6, line 11: what is your reason for deducing that there was feedback between VPS9D1-AS1 and TGF-β, and what is the impact of this feedback.

8. The data volume is sufficient enough to support most of the conclusions, while the logicality of the Results warrants improvement, so does the diagram;

9. Since IFN plays a double role in suppressing or promoting tumor progression, the specific circumstances when employing IFN or target IFN would better be discussed in the Discussion section for better comprehension.

[Editors’ note: further revisions were suggested prior to acceptance, as described below.]

Thank you for resubmitting your work entitled "VPS9D1-AS1 overexpression amplifies intratumoral TGF-β signaling and promotes tumor cell escape from CD8^+^ T cell killing in colorectal cancer" for further consideration by *eLife*. Your revised article has been evaluated by W Kimryn Rathmell (Senior Editor) and a Reviewing Editor.

The manuscript has been improved but there are some remaining issues that need to be addressed, as outlined below:

1) Figure 2A: the artefact previously mentioned (obvious quadrant-type shapes, possibly from stitching scanned images together) have not been addressed. These images should be rescanned. Also, it is unclear how these issues may have impacted the analysis of their data.

2) Figure 2E: this panel has not changed compared to the last time, and contrary to what the authors claim in their reply. And contrarily to what is written on lines 147-149, they are changes in CD8 T cells between VPS9D1-AS1 positive versus negative samples in this cohort.

3) Figure 3 J and K, (and maybe I). The data from the pull-down assays are not fully convincing as the binding of the probes seems to be arbitrary; there is a large difference for RPD1 versus RPD2 in binding RPS3, while these probes are quite similar. Compared to RPD1, RPD2 is missing 1 bp at the 5' end from 1 but has 15 residues added at the 3' end… What is the rationale for this different but comparable design?

4) Along these lines: the short RPD3 construct is the best binder of all the tested compounds, but RPD4 still binds RPS3 and SMAD even having no overlap with RPD3. These data would need to be refined to be publishable and would be better served by being moved to the other study mentioned.

5) Line 256: this sentence claims that TGFb can enter the nucleus to act as a transcription factor. Is this what the authors mean? They should be more explicit in the text. If this is not what they mean, they should rephrase the text accordingly.

6) Figures 5D and 5G: these flow cytometry data need further improvement to be more convincing. E.g. for Figure 5D/upper panels, the gates to indicate IFNAR1 positivity do not seem to be appropriate. Did the authors use an isotype or FMO control for these data to set their gates (this aspect was not mentioned for IFNAR1 in their reply)? Figure 5D: the axes of the flow plots need to be labelled. Also, why the different stainings for CD4 and CD8 T cells in the top and bottom panels? This seems needlessly confusing.

7) Figure 5 Panels E and F: the authors should provide information on the cell types that have been gated on in the graph/axis labels; e.g.: % IFNAR1 of CD4 and CD8, … this would facilitate the understanding of these data. In general, it would be much better and informative to distinguish the expression of these molecules on CD8 and CD4 T cells taken individually (and not together), in particular, if they want to make the point that PD1/PD-L1 interaction has a functional impact on CD8 + T cells in their model. Are the values plotted in panel F directly taken from the FACS plots in panel E (it should be the case to better compare the data)?

8) Figure 5G: to make clearer which cells have which compounds added, I would increase the font a bit, and also add the labels to the bottom panels. Are the authors sure they didn't accidentally switch any of the bottom panels? The percent values indicated in the quadrant do not seem to fit the data plots. Why do the authors detect 2.45% of IFNγ+ OT-1 cells in naïve / unstimulated conditions?

9) Text lines 296-300 ("VPS9D1-AS1 OE reduced the cytotoxicity of activated OT-1 CD8^+^ T cells"): my interpretation of figure 5F differs from the authors. I see the only differences in survival of cancer cells as being due to increased growth of VPS9D1-AS1 OE MC38 cells (as also clearly shown in Figure 6B). This is still a major issue for the paper. I don't think the authors have adequately addressed that the increased tumorigenic potential of VPS9D1-AS1 OE could just be due to enhanced growth, not immune evasion, as posited by the authors. In addition, and as previously mentioned, the effect of PD1 and TGFb blockade are not convincing; the significance appears to come from the higher variation of the data in the condition w/o blockade (and did they add an isotype control there?).

10) Figure 7: There is still no indication in the manuscript methods of the phenotype (and not genetic background) of VPS9D1-AS1 transgenic mice at steady-state / in untreated conditions. In other terms: are these animals entirely normal? i.e. do they show a weight, lifespan, reproductivity, etc that are comparable to their non-transgenic counterparts?

11) Figure 7E: Did 80% of the transgenic mice really die by week 12 of this experiment? And why do in Figure 7L / M transgenic mice in the ASO-NC group start dying at around 10w of AOM/DSS treatment, while there is no death in the ASO-VPS group, and considering the ASO-NC or ASO-VPS treatment only starts from week 14 of AOM/DSS treatment on?

[Editors’ note: further revisions were suggested prior to acceptance, as described below.]

Thank you for resubmitting your work entitled "VPS9D1-AS1 overexpression amplifies intratumoral TGF-β signaling and promotes tumor cell escape from CD8^+^ T cell killing in colorectal cancer" for further consideration by *eLife*. Your revised article has been evaluated by W Kimryn Rathmell (Senior Editor) and a Reviewing Editor.

Several of the previous comments have still not been adequately addressed, but there are still several issues to be resolved, which are listed below. The same numbering is kept to facilitate the reading.

2) Figure 2E: this panel has still not changed compared to the last time, and contrarily to what the authors claim in their reply. Positive sample (please correct spelling in the Figure ) are in red in this Fig2E and show increased numbers of CD8 and CD4 lymphocytes (this is not in line with the text on line 148-149), while FigD indicates increases Ca/STM ratios for CD8 lymphocytes in negative cases (displayed in red).

4) Again: RPD1 and RPD2 are very much comparable, yet their respective ability to binds RPS3 is quite different. In addition, RPD2 entirely overlaps with the sequence of RPD4, yet it binds RPS3 with reduced efficacy. This, together with my previous comments raises concern on the robustness of these data, which the authors did not discuss.

5) There are no definitive date proving TGF-β co-localization in the nucleus; the staining in Figure 3I is difficult to assess. In addition, I could not find clear evidence in the literature that TGF-β can indeed translocate to the nucleus and I would therefore be cautious before making such statement. Did the authors entirely control for the quality of their reagents (specificity of anti-TGF-β antibody, etc.)?

6) In the data newly provided in their reply, the upper row appears to indicate isotypes controls for CD4 and CD3 (and not PD1, as claimed). For the lower row, apparently showing the isotype control, there is a distinct shift in all cell populations between the isotype staining and the staining with the anti-IFNAR1 antibody. This raises concern on the robustness of these data. Possibly, the antibodies need to be titrated and the gating needs to be readjusted. In addition, in Figure 5D, the data show CD4^+^ and CD4- cells (upper row) and CD8^+^ and CD8- cells (lower row). It is formally not correct to assume the CD4- cells are CD8^+^ T cells and vice-versa, since the authors have not combined these antibodies in the same staining panel.

9) Contrarily to what the authors claim, in their reply, there is in Figure 5F not obvious killing between the conditions with MC38+OT-1 without antibody versus MC38+OT-1 and PD1 or TGF-β antibody-mediated blockade (irrespectively of MC38 cells being used as controls or overexpressing VPS) – there is no p value <0.05 indicated for these comparison groups. In other terms, the authors do not compare the right groups to support their statement (lines 299-300). This needs to be addressed and corrected in the manuscript (lines 299-300).

Please improve the figures appropriately to supplement solid evidence, and provide more logical descriptions in the manuscript to fulfil the requirement of *eLife*, which would help accelerate the final decision towards your manuscript.

---

## [Author Response]

Essential revisions:1) The mechanistic study regarding VPS9D1-AS1/TGF-β/ISG was mainly carried out in vitro, while this part shall be tested in cell line-derived xenografts or mouse tumors to support the claims.

In our study, we had applied cell line derived xenograft or allograft tumor models (in vivo) to explore the regarding mechanism for VPS9D1-AS1/TGF-β/ISG. Our study had demonstrated that ASO inhibiting VPS9D1-AS1 decreased the levels of TGFBR1 and TGFβ through HCT116 transplanted xenograft tumor. IFNAR1 was one of an important ISG. In our revised manuscript, we further proved that IFNAR1 played a role on regulating tumor cell proliferation.

2) The data volume is large and the methodology has covered a broad range in the work, while the arrangement of the manuscript could be improved to optimize the presentation and enhance the readability.

According to your and other reviewers’ comments, we shorted our manuscript and made it more readability.

3) Some details of the figures conflict with the statement in the manuscript, and corrections are warranted.

We had carefully read our manuscript and revised these errors in conflicted figures.

Reviewer #1 (Recommendations for the authors):Here are some major concerns:Figure 1/S2: The authors shall explain or discuss why FOXP3 T cells showed a different trend in OUTDO and BJCYH cohorts.

OUTDO cohort only included the cancer tissue samples to subject multiplex multispectral fluorescence immunohistochemistry (mfIHC) analysis, cancer tissues contained tumor parenchyma and tumor stroma, we compared the levels of T cells between cancer tissues and tumor mesenchymal tissues. BJCYH cohort enrolled tumor tissues and normal tissues for mfIHC investigation, and compared the levels of T cells. These reasons caused a different trend for FOXP3 T cells in OUTDO and BJCYH cohorts.

Figure 2: Which cohort(s) was analyzed in 2F? Did OUTDO and BJCYH show the same or different patterns?

In 2F, OUTDO cohort was used to analyze the relationships between VPS and TGFβ signaling genes. We used IHC to stain TGFBR1 to represent TGFβ signaling in BJCYH. The correlation between VPS and TGFBR1 was calculated in ‘Figure 4K’ to illustrate their interrelation with OAS1 and IFNAR1. Both analyses indicated that VPS shown a positive relationship with TGFβ signaling.

Figure 4/S6: 4H-I showed that at 0 hours, OAS1 and IFI27 did not change between CTRL and VPS OE; however, in S6H, OAS1 and IFI27 were much lower in VPS OE, conflicting with 4H-I.

Our results indicate that OAS1 and IFI27 are regulated by VPS, this effect is depending on IFN stimulation (as shown in 4H-I). Once removed IFN in culturing media (S6H), overexpression of VPS would increase both OAS1 and IFI27 expressions. Therefore, these results are no confliction.

Figure 5/S7: In S7A, VPS OE+sgOAS1 (last three lanes) show a substantial OAS1 expression. This made the claims on OAS1 (5B) less convincing. How did VPS KO cells impact IFNAR1 level of T cells in coculture?

In control (CTRL) cells, we demonstrated that sgOAS1 deleted the expression of OAS1 proteins. The S7A result indicate that VPS OE (overexpression) prevent the OAS1 downregulation caused by sgOAS1.

We detected IFNAR1 levels of T cells after co-culturing with VPS OE or sgVPS cells, and proved that IFNAR1 would be effect by these cells. For example, down regulation of IFNAR1 in tumor stroma stimulated CRC development and growth, this finding has been proved by many studies (Inactivation of Interferon Receptor Promotes the Establishment of Immune Privileged Tumor Microenvironment, 2017, Cancer Cell). However, genetic elimination of IFNAR1 in tumor cells enhanced immunological response depended on CD8^+^ T cells and was more susceptibility to T cell-meditated kill (Type I IFN protects cancer cells from CD8^+^ T cell mediated cytotoxicity after radiation, 2019, JCI). Cancer-specific IFNAR1 engagement has been demonstrated to promote cancer stemness and higher expression levels of PDL1 (Cancer-specific type-I interferon receptor signaling promotes cancer stemness and effector CD8^+^ T-cell exhaustion, 2021, Oncoimmunology). Collectively, there are many studies indicate that IFANR1 exert dichotomous function in cancer cells and lymphocytes. Our present study demonstrates that IFNAR1 could be upregulated by VPS in cancer cells, these cells with low IFNAR1 are more susceptible to CD8^+^ T cell killing. On the other hand, we found that cancer-derived IFNAR1 exhibit a negative relationship with CD8^+^ T cells IFNAR1. In this study, we did not dig out the detail mechanism. We hypothesized that IFNAR1 overexpressed in tumor cells regulated many ISGs and then released by tumor cells in TME, which would regulate IFNAR1 in CD8^+^ T cells. This mechanism will be explored in our future study.

Figures 6 and 7: The mechanistic model proposed by in vitro studies (Figures 4 and 5) shall be tested in cell line-derived xenografts or mouse tumors. For example, the IFNAR1 level of VPS OE tumors was examined (7H) but the IFNAR1 level of infiltrated T cells was not. Moreover, CRC patient-derived xenografts shall be used and treated with VPS ASO, which will be more physiologically relevant than the cell line with 600-fold VPS upregulation used in the study.

Our study provides an idea that cancer cell-derived IFNAR1 promotes tumor progression and suppresses anti-tumor immunological reaction in TME. In contrast, these cancer cells with high levels of IFNAR1 are more resistant to CD8^+^ T cell killing, once they contact with CD8^+^ T cells will suppress their IFNAR1 expression. In other word, IFNAR1 mediates the crosstalk between tumor cells and CD8^+^ T cells.

To further validate mechanism regarding VPS promoting tumor through IFNAR1, we downregulated IFNAR1 in CRC cells with VPS overexpression. Our data indicated that IFNAR1 downregulation reversed the promotion for VPS OE on tumor progression.

Moreover, we had proved that VPS was one of drug target and plan to validate this hypothesis using patient-derived xenografts in future study. We thus did not discuss related result in present study.

Reviewer #2 (Recommendations for the authors):My comments are detailed below:(Figure 1)1. Panels A and B (and S1A) are not useful for the conclusions of the paper – the histology is difficult to distinguish and it is not clear of levels (of expression) or frequencies (of positive cells) were quantified in panel B. Those data are better presented in panel D.

We revised this figure to easier understand. Our study applied both RNAscope and qRT-PCR assays to detect the VPS overexpression in CRC patients. In new figure 1, A shows the result of RANscope, we respectively compared their difference expressions in colon cancer and rectal cancer. Panel B shows the data of qRT-PCR, we demonstrated that VPS levels were significantly higher in cancer tissues than their levels in normal tissues. The lower figures of panel B represent the results of paired tissue samples for both BJCYH and OUTDO cohorts.

2. Panel F, for instance, doesn't actually corroborate what is indicated in panel A, as the authors already gate on "high" populations, and there are still some samples with almost no VPS9D1-AS1, this suggests the staining protocol is not a good replacement for qRT PCR.

RNAscope technique enabled direct counting of mRNA molecules in single cells in routine formalin-fixed tissue specimens using bright-field microscopy. RNAscope was considered to superior to real time qRT-PCR because the false negative results obtained from admixtures of many non-malignant cells can be overcome. Thus, our study applied both RNAscope and qRT-PCR to validate the overexpression of VPS using either formalin-fixed CRC tissues or fresh cryopreserved CRC tissues. Panel F tried to mutual confirmation the result derived from two assays. This analysis generally reaches a consensus that VPS is overexpressed in CRC cancer tissues. To more concise, we would like to delete panel F.

3. In the text the authors write that VPS expression doesn't correlate with the expression of genes in the TGFb pathway (pp4, lines 18-20), but do not actually separate by gene expression, but rather by cancer and normal tissue. A better analysis may actually be a xy graph with regression curves.

According this suggestion, *Pearson* correlation analysis was instead of heatmap clustering to investigate their relationships between VPS and TGFβ signaling genes on mRNA levels. This analysis revealed that VPS9D1-AS1 was positive associated with TGFB1 on mRNA levels (as shown in Figure 1F). Because five variables were enrolled these correlation analyses, we used R package ‘corrplot’ to plot correlated coefficients.

(Figure 2)1. There seem to be artefacts or resolution issues in panel A; in panel B, the inlays seem to have different magnifications (size bars are not provided).

mfIHC tissue figures in Panel A were obtained using a Vectra Polaris Imaging System (Perkin Elmer) through 10 X objective lens (100X magnification). Upper pictures show the whole core for each tissue. We enlarged pictures (down) at region of interesting. The size bars were added.

IHC pictures in Panel B were obtained using a white light microscope coupled with scanning system. We show the pictures with 90x magnification for each core. Down pictures were obtained at 400x magnification.

2. It seems data were mixed up data in panel E; in addition, panel D shows Ca/STM ratios while panel E indicates cell numbers, making it difficult to compare the two cohorts. The same applies to Sfig2 panel C versus F.

We sorry for our mistake in panel E, and had revised this error. In panel D, this analysis enrolled cancer tissue samples (OUTDO) using mfIHC method, in contrast, panel E enrolled both cancer and paired normal tissues (BJCHY) using IHC method. mfIHC assay allow us to separate tumor tissues into cancerous tissue and tumor stroma tissue using InForm software. We thus compared the ratios between T cell number in cancerous (Ca) tissues and T cell number in tumor stroma (STM) tissues. For BJCYH, we calculated total number of T cells either whole cancer tissue or paired normal tissue.

3. Panel F needs to be more clearly explained, it is difficult to interpret these data with the explanation given and certain p values are missing. The same applies to SFigure panel G.

According this suggestion, Panel F and S2 panel G had been revised to clearly explain the relationships between TILs and proteins levels involving TGFβ signaling. P values were represented as follows: *P<0.05, **P<0.01, ***P<0.001.

4. Panel A doesn't show what is indicated in the text (cancer vs stromal pp 5 lines 2-3). It shows neg. vs pos. according to the figure and it is difficult to distinguish healthy from tumor tissue.

As mentioned in above explanation, 2A and S2C show the different levels of these T cells between cancer tissues and tumor stroma tissues in OUTDO cohort. 2B and S2D compared the different levels of T cells between cancer tissues and paired normal tissues in BJCYH cohort. The neg. and pos. referred to the expression of VPS in cancer tissues. T cells were compared between CRC patients with neg. VPS in cancer tissues and CRC patients with pos. VPS in cancer tissues (2D and E).

(Figure 3)1. Is SMAD1/5/9 in Figure 3D indeed the top band?

We used an antibody targeting the complex of SMAD1, ~5, ~9. In this picture (actually, 3C), western blotting analysis indicated VPS downregulation decreased the levels of SMAD1/5/9. In Figure 1F, clinical samples analyses indicated that VPS was positively associated with SMAD1 mRNA, other than SMAD9. Besides, we detected the levels of SMAD1, SMAD5, and SMAD9 using specific antibodies for each protein respectively (data not shown), and found SMAD1 was more effect by VPS knockout. Therefore, we inferred that top band should be SMAD1.

2. In panels E and M (and several other western blot data), automated quantification would help support the statements by the authors on increased / lower expression of specific proteins, which are not always convincing as stated.

According this suggestion, we quantified the western blot band for each protein in panels E and M as well as other western blot data in our revised manuscript.

3. L – the oligonucleotide design is not fully clear to me; how can these differences between probes be justified, e.g. between RPD1 and RPD2? Why not compare to the entire VPS9D1-AS1 molecule?

The strategy for designing RNA pull down probes was summarized in S4A. In 3’ end of VPS9D1-AS1 sequence, there are many of repetitive sequence, which increased the difficult to transcript the mRNA of VPS9D1-AS1. In our study, RPD1 and RPD2 were designed to amplify the whole length of VPS9D1-AS1 without repetitive sequence, while RPD3 and RPD4 were used to amplify 400~500bp fragments from 5’ to 3’. The amplified DNA fragments for these probes were then transcript into mRNA using in vitro transcript kit, these mRNAs were further labeled with desthiobiotinylation for RNA pull down assay. This assay indicated that VPS9D1-AS1 bound with RPS3, TGF-β, TGFBR1, SMAD1/5/9 at different sites.

4. O – these data are not convincing.

We had deleted miRNA-related data in our new manuscript according suggestions from you and other reviewers. The mechanism regarding VPS acted as ceRNA will be further discussed in our future study.

Figure 41. In panel E it seems the labels have been swapped.

In panel E we made the wrong labels for IFI27 and OAS1, we are sorry for that. We had revised them.

2. In panel F: there are differences in the two cell lines after VPS-KO; are the labels correct?

HCT116 cells shown higher VPS9D1-AS1 levels than SW480 cells, might be the reasons of genetic background. HCT116 was derived from microsatellite instability CRC patient, in contrast, SW480 was derived from microsatellite stability CRC patient. Our western blot detection demonstrated that pSTAT1 expression was depending on IFN stimulation in SW480 cells. Deleting VPS9D1-AS1 resulted STAT1 disappear in HCT116, but only slightly lowered STAT1 in SW480 cells. Besides, this detection revealed that STAT1/pSTAT1 pathways shown different response to IFN stimulation. To clearly illustrate our viewpoint, we would like to delete the result of SW480.

3. Panel J and references indicated in the text: I could not find in the literature evidence for a nuclear function of TGFb.

Both canonical and noncanonical TGFβ pathway has been well characterized. SMAD4 acted as the transcript factor is the canonical pathway for TGFβ signaling regulating targeting genes. The noncanonical pathway regarding TGFβ activated a series of targeting genes through AKT, mTOR, NF-κB et al. (as mentioned in Reference Derynck, R., Turley, S. J., and Akhurst, R. J. (2021). TGFbeta biology in cancer progression and immunotherapy. Nat Rev Clin Oncol 18, 9-34.). However, the roles regarding tumor-derived TGFβ overexpression are remained to be elucidated. One possible clue for TGFβ controlling OAS1 transcription might be that tumor-derived TGFβ activated AKT, in tumor cells, newly synthesized latent(L)-TGFβ crosslinked to GARP controlled TGFβ signaling (Overcoming TGFβ- mediated immune evasion in cancer, 2022, Nature Reviews Cancer). GARP has been proved to localize in nuclear compartments (GARP as an Immune Regulatory Molecule in the Tumor Microenvironment of Glioblastoma Multiforme, 2019, Int J Mol Sci).

4. In panel K: how were the decimal values for OAS1, IFNAR1, and TGFBR1 determined?

The levels of OAS1, IFNAR1, and TGFBR1 were scored as follows: 1 (0–25%), 2 (26–50%), 3 (51–75%), and 4 (>75%). The intensity of positive staining was classified into 4 scales as follows: 0 (negative), 1 (weak), 2 (moderate), and 3 (strong). The levels were semiquantitatively determined as percentages multiplied by intensity. Therefore, there were decimal values for these proteins.

(Figure 5)1. Panel B: some of the samples seem to have been swapped. Please cross-check.

We recomposed FCM plot panel to show the effects of OAS1 on regulating IFNAR1. Three sgRNAs were used to OAS1 knockout (as shown in S7A). In the sgOAS1 cells, IFNAR1 levels were found to be downregulated. On the other hand, VPS OE enhanced IFNAR1 expression. In 5B, FCM assays were carried out to detect the expressions of IFNAR1. To briefly describe these results, we calculated the difference between CTRL and sgOAS1 cells, as well as VPS OE and VPS OE sgOAS1 cells. All these analyses were based on the results of sgOAS1-1 cells.

2. Panel D: How were patients selected to be either VPS neg. or pos.? This is also relevant due to the apparent intra-tumor heterogeneity of VPS9D1-AS1 expression. FACS plot, VPS positive condition, upper and lower rows: both T cell and non T cell populations are shifted down, respectively up, suggesting a problem or changes in the settings of the flow cytometer instrument. The quality of the flow cytometry data in panel G is also not convincing (several cells are out of the depicted area). Panel E and F: the authors should provide information on the cell types that have been gated on.

Panels D, blood samples and tissues samples were obtained from same patients. RNAscope detected the expression of VPS9D1-AS1 in BJCYH cohort, and defined the patients with VPS neg. or pos. FACS plot had been changed to depict the way to calculate the frequencies of IFNAR1 or PD1 in both CD4 and CD8 T cells.

Panel G, the FACS plots were shown using Pseudocolor type with ‘Smoothing’ model. At new figure, Panel G of the FACS plots were shown without ‘Smoothing’ model.

Panel E and D are the results of FCM detection IFNAR1 and PD1 of CD4 and CD8 T cells. Panel F and G show the data of in vitro OT-1 CD8^+^ T cells killing assays.

3. One concern about the T cell killing (panel F) is that the differences could be due to cell growth changes in the VPS OE cells as seen previously. Is there a way to account for/eliminate that in this setting? Check the units on the y axis of the graph. There is no clear effect of neutralizing antibodies (the conditions including MC38 cells + OT-1 cells should be compared).

Method for T cell killing assays was summarized in ‘supplementary methods’. MC38 CTRL and VPS OE cells were respectively cocultured with OT-1 CD8^+^ T cells for 24 hours. Neutralizing antibodies were mixed in these cultured media. Then, OT-1 CD8^+^ T cells were collected to FCM detection. The remaining MC38 cells were further cultured until to the third days for quantifying cell number. MC38 CTRL was used as control and defined as 1.00, the rest of groups were compared with MC38 CTRL group. The y axis shows the ratios for these analyses.

(Figure 6).1. The murine counterpart of VPS9D1-AS1 is mentioned in the text, but all experiences in vivo are performed using (human) VPS9D1-AS1. The rationale for not using NR045849 in wild-type mice should be better justified.

VPS OE in MC38 cells promoted the expression of TGFβ, TGFBR1, SMAD1, and STAT1, indicated that VPS had similar biological functions between murine cells and human cells. Until to now, no homologous murine gene of VPS was reported. We found NR045849 located similar chromosomal gene location with VPS, but NR045849 was not the homologous gene of VPS. Our western blot analyses indicated that NR045849 had similar biological functions. NR045849 promoted the expression of TGFBR1, SMAD1 and STAT1 in MC38 cells, but not TGFβ, indicated that NR045849 could not instead of VPS. In this context, NR045849 used as the biological control to highlight the biological role of VPS in MC38 cells.

2. Panel B: are there differences in vitro in the growth of control and VPS OE MC38 cells? This should be indicated to better understand the in vivo phenotype of these cells.

CCK8 cell proliferation assays were performed to analyze the growth of MC38 CTRL and VPS OE cells, indicated that VPS significantly promoted MC38 cell proliferation (6B).

3. In general, the flow cytometry data are not convincing. Some cell populations are squished on the top of the graph, there is no indication of what cells are gated on (no gating path or gating strategy indicated). Have isotype controls (or FMO) been used for PD1? The label of the X axis in E should be revised (2 markers on the same axis). If Figure 6D tumor left panel: CD8 numbers are close to 0; therefore, I am not sure how confident one can be about %PD1 in Figure 6F.

We provide the original FCM plots to show the gating strategy. During the FCM detection, an isotype control for each staining antibody was used, included anti-CD3-FITC, anti-CD8-PE, anti-PD1-APC.

Panel D, our FCM detection indicated that little infiltrated CD8^+^ T cell were found in some samples of VPS OE tumor (the CD8^+^ T cell frequencies for VPS OE tumor are 34.7, 2.66, 0.22, 1.5, 5.31, 25.5, 28.8, respectively).

Panel F, the labels for y axis had been changed to CD4/8+ PD1 (%), represent the PD1 frequency in CD4/8+ T cells.

(Figure 7)1. There should be an indication of the phenotype of VPS9D1-AS1 transgenic mice at steady-state; are these animals normal? And is VPS9D1-AS1 indeed only expressed in epithelial cells (this is not clear from FigS9B).

VPS9D1-AS1 transgenic mice were founded based on C57BL/6 background. All mice used in the experiments were backcrossed to C57BL/6 mice for at least 6 generations for maintaining mice at steady-state. This statement has been added in our revised manuscript. In our study, conditional VPS9D1-AS1 transgenic (VPS *^Tg^*) mice were produced through crossed with Cre-villin C57BL/6 mice which allowed VPS9D1-AS1 gene expressed in the intestinal epithelium.

To prove the epithelial location of VPS, we performed RNA FISH assay and found that VPS mainly expressed in the part of intestinal epithelium tissue. The related Figure S9B had been revised and replaced with new pictures.

2. "However, Ifnar1 levels in tumor tissues were increased in VPS Tg mice (Figure 7H), indicating that the tumor-promoting effect of VPS9D1-AS1 OE was dependent on Ifnar1 expression": this is a strong statement or a conclusion that is not supported by the current data. Only a blockade of IFNAR (pharmacologically or genetically) would actually prove this statement.

We agree with reviewer’s comment about IFNAR1. We deleted this statement. We also tried our best to discovery the mechanistic relationships between VPS and IFNAR1 on regulating cell proliferation. Our results shown that VPS OE with IFNAR1 knockdown cells show a lower proliferation rate than VPS OE cells, indicated that IFNAR1 play a role on VPS promoting tumor cell proliferation.

3. What are the units on the Y axis for G? and how were the cells quantified?

Y axis represent the CD8^+^ T cells count. To calculate the CD8^+^ T cells count in mice cancer tissues, immunofluorescence (IF) assays were performed to stain CD8^+^ T cells. For each tissue, 3~5 region-of-interest (ROI) pictures were recorded and manual counted using ImageJ IHC tool, the mean value was represented the CD8^+^ T cell count.

4. Panel H: there seem to be differences in the quality of these stainings. It would be helpful to show healthy intestinal tissue (note that the current magnification does not allow to discern well the malignant tissue). How were IFNARI levels quantified? Note that the same comments also apply to panel N.

We provided the normal tissues pictures for panel H and N. The IFNAR1 levels were quantified using ImageJ IHC tool, the mean values for 3~5 pictures at 200X magnification were calculated for each mouse.

5. I think the author over-interpret the panel K data; again (semi-)quantitative densitometry would help to support their statements.

We had semi-quantitative the western blot results.

The authors do not discuss how the T cells and the VPS9D1-AS1 expressing tumors interact in vivo. Is this also mediated via TGFb (note that this is not shown)? And how do they explain that IFNARI becomes increased on T cells in the tumor micro-environment?

Reviewer proposed a key issue about VPS9D1-AS1. In this study, our in vivo models support the idea that VPS OE prevents CD8^+^ T cell infiltration. IFNAR1 expressed in T cells are related with its antitumor function, many studies had proved this viewpoint. To solve this issue, further studies need to apply TGFβ knockout mice or IFNAR1 knockout mice. We would like to explore their relationships in future studies.

As mentioned in the public review section, the paper could be better served in several ways:1) The paper could be split into two parts. The paper has plenty of data to support the hypothesis of VSP importance in cancer with a less exhaustive look at the binding partners/etc. In particular, the data on downstream miRNAs (Figure 3O, currently not convincing) would benefit from a more in-depth analysis, possibly in a follow-up study that may as well include some of the data presented in Figure 4.

Accordingly, we short our manuscript. Downstream miRNA related findings were deleted and will be report in our next study. We also removed some results from Figure 4 to make our manuscript more concise and easier to understand.

2) The data could be slimmed.

Our manuscript had deleted some data and been slimmed.

A few additional suggestions:– Introduction: the already reported roles of VPS9D1-AS1 for cancer in general and colon cancer, in particular, could be presented in greater detail.

Recent advances about VSP9D1-AS1 had been summarized in the part of ‘Introduction’ and the related references were cited.

– Figure 3A: For non-specialists, it would be helpful to indicate which cells originate from CRC tissue.

The source for these cell lines had been noted in figure legends.

– Figure 3: why swap the colors from F and G? It's confusing.

The color for panels F and G had been revised the consistent color.

– Figure 3: K – the units seem to be wrong in the y axis.

The y axis notes had been corrected. The y axis shows the percentages for intended RIP assay relative to VPS9D1-AS1 in InPut.

– Sfig4G: data on SW480 are too faint to be assessed.

The data related to SW480 had been removed in Figure 4.

– Fig6 and Fig7: these data are not from xenograft experiments since murine cells were injected into mouse recipients (of the same haplotype).

We had revised these related statements in revised manuscript.

– Fig7 panel C (and several of the histology shown): larger magnification should be provided and grading by a pathologist would be beneficial.

The related pictures had been enlarged. All these IHC slices had invited a pathologist to view and confirm.

– Overall, the data are very dense as presented. Their presentation could be improved to enhance readability.

Thanks for your suggestive comments.

Reviewer #3 (Recommendations for the authors):1. Figures to present the expression difference of VPS9D1-AS1 between benign and malignant tissue should be consistent in Figure 1D and 1E for better visualization;

We revised Figure 1D and 1E, to consistent show the difference of VPS9D1-AS1 between benign and malignant tissue in BJCYH and OUTDO cohort, we compared their levels in all normal and cancer tissues, and further selected paired normal and cancer tissues from same patients in two cohort to compare VPS9D1-AS1 levels.

2. The P value between VPS 0 and VPS1 was more prominent than that between VPS 0 and VPS 2-4 in Figure 1F, while the Figure 1G only compares the difference between stroma and VPS 4, please explain the reason;

We tried to compare the VPS9D1-AS1 levels detected using RT-qPCR methods among the RNAscope stained VPS9D1-AS1 with value 0, 1, and 2~4, 0 and 1 represented negative VPS9D1-AS1 expression, 2, 3, 4 represented positive VPS9D1-AS1 expression. This analysis might be confused for reviewers, we deleted this picture in revised manuscript.

3. In Figure 1I, the difference in the expression level of TGFBR1, SMAD1/9, VPS9D1-AS1, and TGF-β seems not significant between the normal and cancer tissue, which is not consistent with the description in the manuscript;

According to your suggestion and other reviewer’s comments, we further analyzed the correlations these genes on mRNA levels using Pearson correlation methods.

4. The quantification of immune cell subsets was presented with two staining methods in Figure 2A and 2B, please specify the reason;

Multiplex multispectral fluorescence immunohistochemistry (mfIHC) assay was carried out to stain immune T cell subsets in OUTDO cohort and IHC assay was performed to show these cells in BJCYH cohort. We intent to cross-validation by using these methods, because the results of mfIHC assay were analyses using Inform software, while immune T cell counts of IHC staining were recorded based on IHC graphs using ImageJ IHC software. Both mfIHC and IHC analyses indicated that VPS9D1-AS1 prevented CD8^+^ T cell tumor infiltrations.

5. The number of CD8^+^ T cells in VPS9D1-AS1 positive tissue overweighs that in VPS9D1-AS1 negative tissue, which is contrary to the description in the manuscript;

Reviewer might refer to Figure 2F. In this figure, the negative and positive VPS were wrongly labeled, we had revised these errors. We are sorry for that.

6. Tumor pictures would be better provided along with the Figure 3G, as was done in Figure 7I;

Accordingly, xenograft tumor pictures for Figure 3G were added.

7. On page6, line 11: what is your reason for deducing that there was feedback between VPS9D1-AS1 and TGF-β, and what is the impact of this feedback.

We found that VPS9D1-AS1 interacted with TGFβ, TGFBR1, SMAD1/5/9 and enhanced their levels. On the other hand, siRNAs targeting these genes inhibited the expression of VPS9D1-AS1. Therefore, we deduced that there is a feedback pathway between VPS9D1-AS1 and TGFβ signaling. This feedback plays a role on maintaining the balance of TGFβ signaling pathway. In tumor cells, high activation of TGFβ signaling pathway prevents T cell killing, high levels of VPS9D1-AS1 help to hold the activation of TGFβ signaling pathway. Once inactivation TGFβ signaling pathway, VPS9D1-AS1 would be downregulated, thus, the inhibition would enable tumor cell to be with low levels of TGFβ signaling pathway and cleared by T cells.

8. The data volume is sufficient enough to support most of the conclusions, while the logicality of the Results warrants improvement, so does the diagram;

Thanks for your positive comments, we had revised to make our manuscript more logical and easier understand.

9. Since IFN plays a double role in suppressing or promoting tumor progression, the specific circumstances when employing IFN or target IFN would better be discussed in the Discussion section for better comprehension.

We mentioned that IFN pathway plays contradictory roles in tumor cells and T lymphocytes. Which indicated that some tumor derived ISG genes overexpression involved in IFN pathway promoting cancer progression, such as OAS1. On the other hand, ISG gene expressed by T lymphocytes, such as IFN-γ and IFN-α/β，inhibited tumor progression. We had discussed these roles in the part of ‘introduction’ (line 9-20, page 3) and ‘discussion’ (line 8-24, page 12) in our manuscript.

[Editors’ note: further revisions were suggested prior to acceptance, as described below.]

The manuscript has been improved but there are some remaining issues that need to be addressed, as outlined below:1) Figure 2A: the artefact previously mentioned (obvious quadrant-type shapes, possibly from stitching scanned images together) have not been addressed. These images should be rescanned. Also, it is unclear how these issues may have impacted the analysis of their data.

We rescanned these tissue sample using Perkin Elmer Vetra microscope to avoid quadrant-type shapes and updated represent images in Figure 2A. Inform software was used to our analysis about CD4. CD8, and FOXP3 T cells in CRC tissues. To ensure the accuracy of our data, the CD4. CD8, and FOXP3 T cell frequencies for each tissue was analyzed at one time, this analysis enrolled all regions of each scanned picture.

2) Figure 2E: this panel has not changed compared to the last time, and contrary to what the authors claim in their reply. And contrarily to what is written on lines 147-149, they are changes in CD8 T cells between VPS9D1-AS1 positive versus negative samples in this cohort.

We carefully considered this issue proposed by reviewer. We compared our revised manuscript version with our first submitted manuscript, the errors for Figure 2E had been corrected (we wrongly labeled VPS negative and positive). In Figure 2E, our results indicated that patients with VPS positive expression shown lower levels of infiltrated CD8^+^ T cells and CD4^+^ T cells than these patients with negative expression of VPS. This result is consistent with Figure 2D. The color for Figure 2D were revised to consist with Figure 2E for VPS negative and positive. But we did not find the contrarily between our written on lines 147-149 and Figure 2E. Besides, we rephrased this sentence to more rational raise our viewpoint.

3) Figure 3 J and K, (and maybe I). The data from the pull-down assays are not fully convincing as the binding of the probes seems to be arbitrary; there is a large difference for RPD1 versus RPD2 in binding RPS3, while these probes are quite similar. Compared to RPD1, RPD2 is missing 1 bp at the 5' end from 1 but has 15 residues added at the 3' end… What is the rationale for this different but comparable design?

Figure 3J is RIP assay, use antibodies to bind with VPS9D1-AS1. Figure 3K is RNA pulldown assay. The probes used in RNA pulldown assay were four transcripts of VPS9D1-AS1, which were transcribed using an in vitro RNA Synthesis Kit. Then, the RNA products were labeled with desthiobiotinylation. After that, these labeled VPS9D1-AS1 sequences were mixed with cell lysis. The binding proteins were tested using western blotting at last. High repeat sequences of VPS9D1-AS1 result that the transcription of full length VPS9D1-AS1 is very difficult in vitro. To transcribe RNA transcripts of VPS9D1-AS1 from cDNA template, two paired primers (RPD1 and RPD2, 1bp is different for upper primer) were used to transcribe the long probes, which is similar to the full length of VPS9D1-AS1. RPD1 and RPD2 share some same sequence with RPD3.

4) Along these lines: the short RPD3 construct is the best binder of all the tested compounds, but RPD4 still binds RPS3 and SMAD even having no overlap with RPD3. These data would need to be refined to be publishable and would be better served by being moved to the other study mentioned.

Follow up last question, the probes for RPD3 and PRD4 were design to transcribed a short RNA sequence of VPS9D1-AS1 to identify the difference binding ability between VPS9D1-AS1 and targeted proteins. Four probes can bind with RPS3. RPD3 shown higher binding ability with RPS3, TGF-β and SMAD1/5/9. RPD2 show a higher binding ability with TGFBR1. These results indicated that VPS9D1-AS1 located in ribosome of cancer cell by interacting with RPS3 and then regulated the transcriptions of TGF-β, TGFBR1 and SMAD1/5/9 through different sequence. In our study, Figure 3J was designed to proteins binding with VPS9D1-AS1, meanwhile, Figure 3K was designed to VPS9D1-AS1 binding with proteins, each assay complementing each other.

5) Line 256: this sentence claims that TGFb can enter the nucleus to act as a transcription factor. Is this what the authors mean? They should be more explicit in the text. If this is not what they mean, they should rephrase the text accordingly.

According to your suggestion, we rephrase this sentence. The reference (Derynck et al., 2021, TGFbeta biology in cancer progression and immunotherapy. Nat Rev Clin. ) indicated that TGFβ-induced activation of ERK–MAPK, AKT and NF-κB signaling might initiate from the distinct receptor complexes. Our study tried to reveal both SMAD4 and TGF-β acting as regulators of OAS1 gene. SMAD4 has been prove to act as the transcription factor for many genes. Our study indicated that TGF-β might be of direct or indirect transcription factor to control OAS1 expression. IF staining (Figure 3I) indicate that TGF-β can be found to co-locate with nucleus of HCT116 cells. Our ChIP assay further demonstrated that TGF-β-antibody could IP the promotor region of OAS1. This might be refer to some unknown mechanism for TGF-β launching intracellular signaling in cancer cells.

6) Figures 5D and 5G: these flow cytometry data need further improvement to be more convincing. E.g. for Figure 5D/upper panels, the gates to indicate IFNAR1 positivity do not seem to be appropriate. Did the authors use an isotype or FMO control for these data to set their gates (this aspect was not mentioned for IFNAR1 in their reply)? Figure 5D: the axes of the flow plots need to be labelled. Also, why the different stainings for CD4 and CD8 T cells in the top and bottom panels? This seems needlessly confusing.

In the experiments of Figure 5D, isotypes for IFNAR1 and PD1 was carried out to quantitative detect the levels of IFNAR1 and PD1 in blood T lymphocytes. The positive expressed IFNAR1 as well as PD1 were determined according to isotypes for each FCM experiment (as shown in Author response image 1).

The axes labels for Figure 5D had been added.

The antibody panel for quantitative detecting IFNAR1 included CD3-FITC, CD4-APC, and IFNAR1-PE in CD4 and CD8 cells. While, CD3-FITC, CD8-PE, and PD1-APC antibodies consist the panel to quantitatively detect PD1 in CD4 and CD8 cells. Therefore, the FCM dots for these experiments were different.

**Author response image 1. sa2fig1:** Isotype for IFNAR1 in CD4 and CD8 T cells.

7) Figure 5 Panels E and F: the authors should provide information on the cell types that have been gated on in the graph/axis labels; e.g.: % IFNAR1 of CD4 and CD8, … this would facilitate the understanding of these data. In general, it would be much better and informative to distinguish the expression of these molecules on CD8 and CD4 T cells taken individually (and not together), in particular, if they want to make the point that PD1/PD-L1 interaction has a functional impact on CD8 + T cells in their model. Are the values plotted in panel F directly taken from the FACS plots in panel E (it should be the case to better compare the data)?

According to your suggestion, we had revised the labels for Figure 5E.

In Figure 5F, we employed an in vitro Cytotoxicity Assay to test the cytotoxicity CD8^+^ OT-I cells on killing MC38 cancer cells. This is one kinds of pattern recognition model, which MC38 cells were transfected by OVA vector. OVA is the antigen can be recognized by CD8^+^ OT-I cells. In panel F (right), the remaining cell number were calculated and compared their ratios (relative to group MC38-CTRL). The results for Panel F are not taken from FACS plots.

8) Figure 5G: to make clearer which cells have which compounds added, I would increase the font a bit, and also add the labels to the bottom panels. Are the authors sure they didn't accidentally switch any of the bottom panels? The percent values indicated in the quadrant do not seem to fit the data plots. Why do the authors detect 2.45% of IFNγ+ OT-1 cells in naïve / unstimulated conditions?

Accordingly, the fonts for Figure 5G were enlarged. We did not switch the bottom panels. In this FCM detection experiment, OT-I cells were collected and staining with CD8-PE and IFN-γ-APC antibodies. We used isotype for CD8-PE to gate OT-I cells, to distinguish mixed MC38 cells in cultured media.

The right panel of 5G analyzed the IFN-γ levels (relative to untreated OT-I) in different groups.

The OT-1 cells were separated from OT-1 mice spleen and subjected to the media supplied IL-1 and CD3/CD28 antibodies to prime. After this treatment, the expression of IFN-γ should be activated, therefore, we could detect IFN-γ in these OT-I cells.

9) Text lines 296-300 ("VPS9D1-AS1 OE reduced the cytotoxicity of activated OT-1 CD8^+^ T cells"): my interpretation of figure 5F differs from the authors. I see the only differences in survival of cancer cells as being due to increased growth of VPS9D1-AS1 OE MC38 cells (as also clearly shown in Figure 6B). This is still a major issue for the paper. I don't think the authors have adequately addressed that the increased tumorigenic potential of VPS9D1-AS1 OE could just be due to enhanced growth, not immune evasion, as posited by the authors. In addition, and as previously mentioned, the effect of PD1 and TGFb blockade are not convincing; the significance appears to come from the higher variation of the data in the condition w/o blockade (and did they add an isotype control there?).

As shown in Figure 5F, this T cell cytotoxicity assay was used to determine the inhibited ability of CD8^+^ OT-I cells in killing tumor cell with supplied TGF-β or PD-1 antibodies. In Figure 5F (left), we really observed that CD8^+^ T OT-I cells killed MC38 CTRL-OVA cells as well as MC38-VPS OE-OVA cells. However, there is no significantly difference on proliferated rates between MC38 CTRL-OVA and MC38-VPS OE-OVA cells after treatment with CD8^+^ T OT-I. In view of this, we agree with reviewer’ viewpoint and revised this sentence.

We repeated three times for T cell cytotoxicity assay and observed that PD1 antibody and TGF-β antibody supplied in the cultured media significantly enhanced CD8^+^ T OT-I cells for each experiment. Besides, isotype control was applied to gate CD8^+^ OT-I cells in Figure 6G.

10) Figure 7: There is still no indication in the manuscript methods of the phenotype (and not genetic background) of VPS9D1-AS1 transgenic mice at steady-state / in untreated conditions. In other terms: are these animals entirely normal? i.e. do they show a weight, lifespan, reproductivity, etc that are comparable to their non-transgenic counterparts?

In our mice models, VPS9D1-AS1 were conditional knock in murine genome, which did not cause any changes in mouse traits (such as lifespan and bodyweight). According to this suggestion, we had described this indication in our manuscript.

11) Figure 7E: Did 80% of the transgenic mice really die by week 12 of this experiment? And why do in Figure 7L / M transgenic mice in the ASO-NC group start dying at around 10w of AOM/DSS treatment, while there is no death in the ASO-VPS group, and considering the ASO-NC or ASO-VPS treatment only starts from week 14 of AOM/DSS treatment on?

In Figure 7E, three cycles of DSS after one week of AOM treatment were treated WT and VPS^tg^ mice. In contrast, one cycle of DSS treated mice after one week of AOM treatment in Figure 7L. Therefore, more DSS treatment lead higher death rate in Figure 7E than Figure 7L/M. To view the therapeutic effect of ASO-VPS, we used one cycle of DSS induced tumor in Figure 7L/M. After DSS treatment for one week, ASO-VPS and ASO-NC drugs were applied to treat mice. This would be the reason for ASO-NC group start dying at 10^th^ week.

[Editors’ note: further revisions were suggested prior to acceptance, as described below.]

Several of the previous comments have still not been adequately addressed, but there are still several issues to be resolved, which are listed below. The same numbering is kept to facilitate the reading.2) Figure 2E: this panel has still not changed compared to the last time, and contrarily to what the authors claim in their reply. Positive sample (please correct spelling in the Figure ) are in red in this Fig2E and show increased numbers of CD8 and CD4 lymphocytes (this is not in line with the text on line 148-149), while FigD indicates increases Ca/STM ratios for CD8 lymphocytes in negative cases (displayed in red).

We deeply sorry for our mistake in Figure 2E and thanks for your kindly comment. At this time, we accordingly revised these figures. Previous figure 2E wrongly labeled ‘positive’ (red) and ‘negative’ (blue) group. We fixed these errors. The labels for Figure 2D were correct, the Ca/STM ratios for CD4/8 lymphocytes were increased in patient with negative VPS9D1-AS1 expression.

4) Again: RPD1 and RPD2 are very much comparable, yet their respective ability to binds RPS3 is quite different. In addition, RPD2 entirely overlaps with the sequence of RPD4, yet it binds RPS3 with reduced efficacy. This, together with my previous comments raises concern on the robustness of these data, which the authors did not discuss.

We firstly response to reviewer previous comment. In view that, the binding VPS9D1-AS1 with TGF-β, TGFBR1, and SMAD1/5/9 had been proved by several assays in our study, as shown in Figure 3I and J. We thus revised RNA-pull down related data and deleted the results of TGF-β, TGFBR1, and SMAD1. According reviewer suggestion, these issues should be moved to another research to be addressed. All related description and figure legend had been consistent revised. Among of all these detecting binding proteins for VPS9D1-AS1, the RNA-pull down assay for RPS3 was better than other protein. In this revised manuscript, we tried to preserve this result for VPS9D1-AS1 binding with RPS3, this would help to hint the subcellular location for VPS9D1-AS1.

About RPD probes concern, as be pointed out by reviewer, four designed probes show different binding ability with VPS9D1-AS1 in our RNA-pull down assay. We thought that the specific repetitive sequences (from 1298bp to 1753bp, total 279 bp) in the 3’ end of VPS9D1-AS1 play an important role during binding with RPS3 and other proteins. The most important difference between RPD1 and RPD2 are the sequences in 3’end, RPD2 contained 13 bp repetitive sequences. On the other hand, RPD3 is the 5’ end transcript of VPS9D1-AS1, shown the highest binding affinity with RPS3, indicated that VPS9D1-AS1 bound with RPS1 through 5’ end of RNA chain. Although RPD3 shared same sequence with RPD1 and RPD2, did not contained repetitive sequences, shown higher binding ability than RPD2, but lower than RPD1 and RPD3, which further indicated the binding mainly occurring in the 5’ end.

5) There are no definitive date proving TGF-β co-localization in the nucleus; the staining in Figure 3I is difficult to assess. In addition, I could not find clear evidence in the literature that TGF-β can indeed translocate to the nucleus and I would therefore be cautious before making such statement. Did the authors entirely control for the quality of their reagents (specificity of anti-TGF-β antibody, etc.)?

According this suggestion, we revised the related statement in our manuscript (line 256~262, page 8~9). Figure 4J and legend were also consistently revised to delete the results of TGF-β. The issue for TGF-β acting as transcription factor might be discussed in future study.

6) In the data newly provided in their reply, the upper row appears to indicate isotypes controls for CD4 and CD3 (and not PD1, as claimed). For the lower row, apparently showing the isotype control, there is a distinct shift in all cell populations between the isotype staining and the staining with the anti-IFNAR1 antibody. This raises concern on the robustness of these data. Possibly, the antibodies need to be titrated and the gating needs to be readjusted. In addition, in Figure 5D, the data show CD4^+^ and CD4- cells (upper row) and CD8^+^ and CD8- cells (lower row). It is formally not correct to assume the CD4- cells are CD8^+^ T cells and vice-versa, since the authors have not combined these antibodies in the same staining panel.

The strategies for gating IFNAR1 and PD1 in T cells were shown as follow picture. Our previous response indicated the method for gating IFNAR1. The lower row represented an isotype sample with staining ISO-PE antibody. On the other hand, right graph shown the CD3/CD4 staining cells. This might be the reason for the shift in cell populations between lower graphs. We further regrouped these FCM graphs and tried to show gating strategies. We used isotype as the negative controls for gating positive IFNAR1 and PD1 cells to represent their levels in CD4 and CD8 T cells. As our mentioned in our previous response, we used CD3/CD4 antibodies and CD3/CD8 antibodies to discern CD4 and CD8 cells, this method allowed us to reduce an antibody for each panel, this might be not rigorous enough. However, CD3/CD4/IFNAR1 antibodies staining could represent the levels of IFNAR1 in CD4 cells, while, CD3/CD8/PD1 antibodies staining could represent the levels of PD1 in CD8 cells. Therefore, we revised our results in Figure 5 D and E. The related statement had also been revised in manuscript.

9) Contrarily to what the authors claim, in their reply, there is in Figure 5F not obvious killing between the conditions with MC38+OT-1 without antibody versus MC38+OT-1 and PD1 or TGF-β antibody-mediated blockade (irrespectively of MC38 cells being used as controls or overexpressing VPS) – there is no p value <0.05 indicated for these comparison groups. In other terms, the authors do not compare the right groups to support their statement (lines 299-300). This needs to be addressed and corrected in the manuscript (lines 299-300).

Current comment help us to more precise understand reviewer’s intent. We labeled the p values for comparison among VPS OE, VPS OE+OT-1, VPS OE+OT-1+TGF-β^ab^, and VPS OE+OT-1+PD1^ab^ groups. TGF-β^ab^ and PD1^ab^ supplied in media lowered MC38 VPS OE cell proliferation, but there are no statistically significant difference. We thus revised our statement in manuscript.